# COOPERATIVE VARIANCE ESTIMATION AND BAYESIAN NEURAL NETWORKS DISENTANGLE ALEATORIC AND EPISTEMIC UNCERTAINTIES

## ABSTRACT

Real-world data contains aleatoric uncertainty – irreducible noise arising from imperfect measurements or from incomplete knowledge about the data generation process. Mean variance estimation (MVE) networks can learn this type of uncertainty but require ad-hoc regularization strategies to avoid overfitting and are unable to predict epistemic uncertainty (model uncertainty). Conversely, Bayesian neural networks predict epistemic uncertainty but are notoriously difficult to train due to the approximate nature of Bayesian inference. We propose to cooperatively train a variance estimation network with a Bayesian neural network and empirically demonstrate that the resulting model disentangles aleatoric and epistemic uncertainties while improving the mean estimation. We demonstrate the effectiveness and scalability of this method across a diverse range of datasets, including a time-dependent heteroscedastic regression dataset we created where the aleatoric uncertainty is known, used to assess estimation accuracy. The proposed method is straightforward to implement, robust, and adaptable to various model architectures.

## 1 INTRODUCTION

Non-probabilistic neural networks that only estimate the mean (expected value) tend to be overconfident and vulnerable to adversarial attacks (Guo et al., 2017a; 2019). Quantifying aleatoric (or data) uncertainty alleviates these issues by characterizing noise (Skafte et al., 2019; Seitzer et al., 2022). Estimating epistemic (or model) uncertainty enables active learning and risk-sensitive decision-making (Kendall and Gal, 2017; Depeweg et al., 2018). Consequently, except in cases with negligible or homoscedastic data noise, simultaneously predicting aleatoric and epistemic uncertainties is essential for a wide range of safety-critical applications (Hüllermeier and Waegeman, 2021). In such cases, an outcome with good mean performance but large aleatoric uncertainty may be undesired (Lakshminarayanan et al., 2017). Therefore, the principle of reducing epistemic uncertainty behind active learning or decision-making needs to be balanced by the respective prediction of aleatoric uncertainty. This is particularly important when the aleatoric uncertainty is heteroscedastic (input-dependent) due to imperfect measurements, environmental variability, and other factors (Smith et al., 2024).

**Summary of contributions.** We propose a cooperative learning strategy for uncertainty disentanglement based on sequential training of (1) a mean network, (2) a variance network, and (3) a Bayesian neural network. Figure 1 illustrates the method for one-dimensional heteroscedastic regression, briefly describing it at the figure caption.

## 2 RELATED WORK

### 2.1 ALEATORIC UNCERTAINTY

Aleatoric uncertainty can be estimated by parametric or nonparametric models. The latter do not explicitly define the likelihood function and instead focus on learning to sample from the data distribution (Mohamed and Lakshminarayanan, 2016). Their ability to estimate nontrivial aleatoric

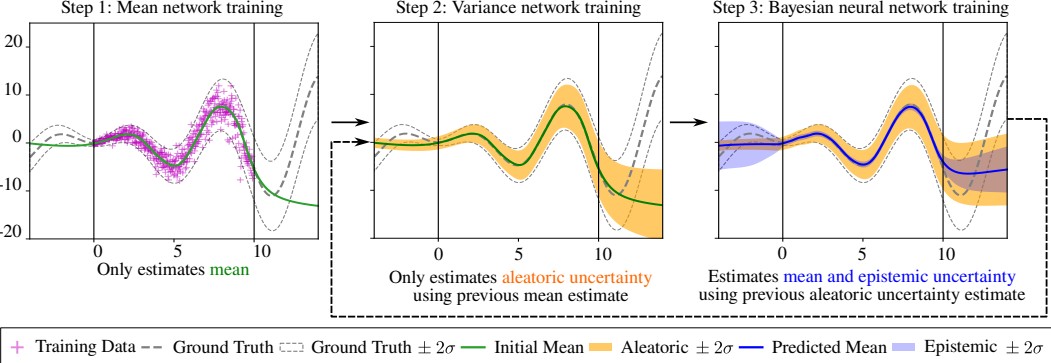

Figure 1: **Illustration of the proposed cooperative training strategy leading to Variance estimation Bayesian Neural Networks (VeBNNs).** The left figure shows the unseen ground truth mean as well as the respective training data. The method starts by training the mean network to only estimate the mean (green solid line in Step 1). Then, without updating the mean estimate, a variance network is trained to only predict aleatoric uncertainty (orange credible interval in Step 2). Subsequently, considering this aleatoric uncertainty estimation, a Bayesian neural network is trained to obtain an updated mean and corresponding epistemic uncertainty (solid blue line for the new mean, and shaded blue credible interval for the epistemic uncertainty in Step 3). If needed, the method can iterate between steps 3 and 2 to improve the disentanglement of uncertainties. Note the disentanglement of uncertainties together with the improvement of the mean estimation away from the data support ($x < 0$ and $x > 10$ in Step 3).

uncertainty comes at the cost of training difficulties and sampling inefficiencies (Sensoy et al., 2020; Harakeh et al., 2023). Therefore, parametric models are more common. They assume a parameterized observation distribution, usually a Gaussian as in Mean Variance Estimation (MVE) networks (Nix and Weigend, 1994), and learn the corresponding parameters by minimizing the Negative Log-Likelihood (NLL) loss with associated regularization. Similar parametric models have been developed, replacing the Gaussian distribution with other distributions (Meyer and Thakurdesai, 2020). However, training these models can be challenging. MVE networks have been observed to lead to good mean but overconfident variance estimations in regions within the data support, and exhibit generalization issues outside these regions (Skafte et al., 2019). As analyzed by different authors (Skafte et al., 2019; Seitzer et al., 2022; Immer et al., 2023; Sluijterman et al., 2024), these issues result from the loss gradients having very different magnitudes when calculated with respect to the mean or the aleatoric variance, as seen in Equation (3), which creates imbalances when minimizing the NLL.

Different solutions have been proposed to improve the parametric estimation of aleatoric uncertainty, including a Bayesian treatment of variance (Stirn and Knowles, 2020), calibration by distribution matching via maximum mean discrepancy loss (Cui et al., 2020), or modifying the loss function to include an additional balance hyperparameter (Seitzer et al., 2022). However, Sluijterman et al. (2024) demonstrated that training an MVE network that simultaneously predicts mean and variance leads to significantly worse predictions for both estimations compared to separately training, even when considering the above-mentioned modified losses (we independently reached the same conclusion while conducting our work; see Appendix A.1). Nevertheless, we note that parametric deterministic models are insufficient if they only estimate mean and aleatoric uncertainty, without accounting for epistemic uncertainty. This is also visible in Figure 1 (Step 2), as the model has an overconfident mean outside the data support (see $x > 10$). Furthermore, the inability to estimate epistemic uncertainty is problematic in itself, as discussed in Section 1.

## 2.2 EPISTEMIC UNCERTAINTY

Epistemic uncertainty is usually estimated by probabilistic models that impose a prior distribution on the model parameters (Abdar et al., 2021). Instead of finding point estimates as in deterministic models, they predict a posterior predictive distribution (PPD). Unfortunately, accurate and computationally tractable determination of the PPD is challenging in most cases, except for a few models like Gaussian processes, where inference can be done exactly under strict assumptions and with limited data scalability (Rasmussen and Williams, 2005). Bayesian neural networks (BNNs) are more

scalable, but the accuracy and scalability are strongly dependent on the type of Bayesian inference (Wilson and Izmailov, 2020; Wenzel et al., 2020; Izmailov et al., 2021).

Bayesian inference is commonly done by Markov Chain Monte Carlo (MCMC) sampling methods (Neal, 1995; Welling and Teh, 2011; Li et al., 2016) or Variational Inference (VI) methods that are faster to train but less accurate (Graves, 2011; Blundell et al., 2015). Avoiding formal Bayesian inference is also possible by adopting ensemble methods such as Monte Carlo (MC) Dropout (Gal and Ghahramani, 2016) and deep ensembles (Lakshminarayanan et al., 2017; Fort et al., 2020). Although not strictly Bayesian, MC-Dropout can be interpreted as a Variational Bayesian approximation that has additional scalability but even lower accuracy (no free lunch).

The practical difficulties of training BNNs have limited their widespread use. Therefore, they are often trained by disregarding aleatoric uncertainty or assuming homoscedastic noise, which can be set as a hyperparameter but is impractical for complex problems (Abdar et al., 2021). Instead, the end-to-end training (also referred to as *second order* uncertainty quantification) that trains an MVE network by MC-Dropout (Kendall and Gal, 2017) or deep ensembling (Wang et al., 2025) is reported to approximately disentangle aleatoric and epistemic uncertainties for heteroscedastic regression and classification. However, the essential challenges faced when training MVE networks are not solved by ensembling them, and the lack of accuracy associated with MC-Dropout raises questions about their ability to make high-quality predictions and truly disentangle epistemic and aleatoric uncertainties (Valdenegro-Toro and Mori, 2022; Mucsányi et al., 2024). This has motivated other authors to propose different solutions. A non-Bayesian solution to separate uncertainties was proposed by introducing a high-order evidential distribution, i.e., considering priors over the likelihood function instead of over network weights (Amini et al., 2020). Another proposal has been to use a natural reparameterization combined with an approximate Laplace expansion to estimate epistemic uncertainty (Immer et al., 2023). Still, a recent investigation (Mucsányi et al., 2024) suggests that no current method achieves reliable uncertainty disentanglement.

## 3 METHODOLOGY

### 3.1 PRELIMINARIES

Consider a dataset $\mathcal{D} = \{\mathbf{x}_n, y_n\}_{n=1}^N$ with *i.i.d.* data points, where $\mathbf{x}_n \in \mathbb{R}^d$ represents the input features and $y_n \in \mathbb{R}$ the corresponding outputs[1]. The heteroscedastic regression problem can be formulated as:

$$y = f(\mathbf{x}) + \epsilon(\mathbf{x}) \tag{1}$$

where $f(\mathbf{x})$ denotes the underlying noiseless ground truth function (expected mean), and $\epsilon(\mathbf{x})$ is the corresponding heteroscedastic noise (aleatoric uncertainty). If the noise is Gaussian, then $\epsilon(\mathbf{x}) \sim \mathcal{N}(0, s^2(\mathbf{x}))$ where $s^2(\mathbf{x})$ represents its ground truth input-dependent variance.

### 3.2 CHALLENGES OF JOINT TRAINING OF MEAN VARIANCE NETWORK

The NLL loss used for training MVE networks resulting from a Gaussian observation distribution with heteroscedastic aleatoric uncertainty is:

$$\mathcal{L}_1(\boldsymbol{\theta}, \boldsymbol{\phi}) = \sum_{n=1}^N \left[ \frac{(y_n - \mu(\mathbf{x}_n; \boldsymbol{\theta}))^2}{2\sigma_a^2(\mathbf{x}_n; \boldsymbol{\phi})} + \frac{1}{2} \log\left(\sigma_a^2(\mathbf{x}_n; \boldsymbol{\phi})\right) \right] \tag{2}$$

where $\sigma_a^2(\mathbf{x}; \boldsymbol{\phi}) > 0$ is the parameterized aleatoric variance, and $\mu(\mathbf{x}; \boldsymbol{\theta})$ the mean. The derivatives with respect to $\mu(\mathbf{x}; \boldsymbol{\theta})$ and $\sigma_a^2(\mathbf{x}; \boldsymbol{\phi})$ become:

$$\nabla_\mu \mathcal{L}_1 = \sum_{n=1}^N \left( \frac{\mu(\mathbf{x}_n; \boldsymbol{\theta}) - y_n}{\sigma_a^2(\mathbf{x}_n; \boldsymbol{\phi})} \right), \; \nabla_{\sigma_a^2} \mathcal{L}_1 = \sum_{n=1}^N \left( \frac{\sigma_a^2(\mathbf{x}_n; \boldsymbol{\phi}) - (y_n - \mu(\mathbf{x}_n; \boldsymbol{\theta}))^2}{2\left(\sigma_a^2(\mathbf{x}_n; \boldsymbol{\phi})\right)^2} \right). \tag{3}$$

---

[1]The equations become simpler when writing for one-dimensional outputs, but this article includes examples with multi-dimensional outputs $\mathbf{y}_n \in \mathbb{R}^m$, as shown later.

According to Equation (3), the gradient with respect to $\sigma_a^2(\mathbf{x}; \phi)$ has a high-order term in the denominator that makes the optimization more challenging and causes an imbalance when minimizing the loss. Due to space constraints, we show in Appendix A.2 that the loss of jointly trained MVE networks is not convex with respect to their outputs (the mean and aleatoric variance). This explains the above-mentioned optimization challenges that have been empirically observed in the literature and that we also observe in our work. As a result, joint (or end-to-end) training of MVE networks can lead to unexpected outcomes, for example: (1) if an MVE network overfits the response, i.e., $(y_n - \mu(\mathbf{x}_n; \boldsymbol{\theta}))^2 \to 0$, $\sigma_a^2(\mathbf{x}_n; \phi) \to 0$, the loss function $\mathcal{L}_1$ may become undefined; (2) if the MVE network leads to large variance estimates when quantifying aleatoric uncertainty, then $\sigma_a^2(\mathbf{x}_n; \phi) \to \infty$ and the gradients with respect to $\sigma_a^2(\mathbf{x}_n; \phi)$ vanish. The saddle shape of the loss function translates into low-quality parameter point estimates or slower convergence, as training may become trapped in degenerate regions (see Appendix A). While balancing the loss is possible, as proposed in (Seitzer et al., 2022) by the $\beta$-NLL loss, this introduces additional hyperparameters that need to be optimized for each problem. Furthermore, conducting Bayesian inference for MVE networks such that we simultaneously predict aleatoric and epistemic uncertainties should only make these issues more acute, as evidenced in the literature (Mucsányi et al., 2024).

### 3.3 VEBNN: COOPERATIVE VARIANCE ESTIMATION BAYESIAN NEURAL NETWORKS

We propose a sequential training strategy (see Algorithm 1) that starts with training a mean network, then a variance estimation network that predicts aleatoric uncertainty, followed by a BNN that updates both mean and epistemic uncertainty for the previously determined aleatoric uncertainty. By separating the roles of each network and ensuring their cooperative training, we demonstrate that the resulting BNN can learn and improve all three estimates. Importantly, training each network separately is easier, and we show that inference of the BNN is also facilitated due to the presence of a good estimate of aleatoric uncertainty (that is not being learned at that stage). Appendix A.3 includes theoretical arguments in favor of our cooperative training strategy, also explaining why there is convergence for a growing number of iterations $K$.

---

**Algorithm 1:** Cooperative VeBNN training

**Data:** mean network $\mu(\mathbf{x}; \boldsymbol{\theta})$, variance network $\sigma_a^2(\mathbf{x}; \phi)$, dataset $\mathcal{D} = \{\mathbf{X}, \mathbf{y}\}$, number of iterations $K$

**Step 1: mean network training:** Minimize Equation (2) for constant $\sigma_a^2$ to find point estimate of $\hat{\mu}(\mathbf{x}; \hat{\boldsymbol{\theta}})$, i.e., minimize Equation (16).

**for** $i = 1$ **to** $K$ **do**

  **Step 2: variance network training (Aleatoric uncertainty):** Minimize Equation (6) for fixed mean from Step 1 or Step 3 to find point estimate $\hat{\sigma}_a^2(\mathbf{x}; \hat{\phi})$ from Equation (8).

  **Step 3: Bayesian network training (Epistemic uncertainty):** Sample posterior from Equation (9) to determine mean and epistemic variance estimates for fixed aleatoric variance found in Step 2 via Equation (8); and compute the log marginal likelihood: $\texttt{LMglk}[i] = \log \mathbb{E}_{p(\boldsymbol{\theta}|\mathcal{D})} \left[ p(\mathbf{y} \mid \boldsymbol{\theta}^{(i)}, \phi^{(i)}) \right]$

**end**

Identify the optimal parameters $\boldsymbol{\theta}^*$ and $\phi^*$ by $i^* = \arg\max_i \texttt{LMglk}[i]$

---

The proposed Algorithm 1 starts by considering constant aleatoric uncertainty, i.e. $\sigma_a^2(\mathbf{x}; \phi) = $ constant, and conventionally trains the mean network by finding the maximum a posteriori (MAP) estimate of the parameters $\boldsymbol{\theta}$ of Equation (2), hence determining only the mean.

#### 3.3.1 VARIANCE ESTIMATION (VE) NETWORK TRAINING

Once the mean is obtained, we then train the variance estimation (Ve) network for this fixed mean. There are, however, multiple important details that facilitate training of this network (Step 2 in Algorithm 1). The variance estimation network does not directly output $\sigma_a^2(\mathbf{x}; \phi)$, and so its parameters are not directly determined by minimizing Equation (2) and keeping the mean fixed. Instead, the variance network outputs the residual $r = (\mu(\mathbf{x}; \boldsymbol{\theta}) - y)^2$, which follows a Gamma distribution because $y$ is Gaussian.

**Assumption 3.1.** $(\mu(\mathbf{x}; \boldsymbol{\theta}) - f(\mathbf{x}))^2$ *is finite and tends to 0 when* $N \to \infty$. *This follows from assuming unbiased or consistent estimations for the ground truth* $f(\mathbf{x})$, *regardless of noise.*

**Lemma 3.1.** *Given Assumption 3.1 and assuming that $y$ follows a Gaussian distribution, the squared residual $r = (\mu(x;\theta) - y)^2$ follows a Gamma distribution.*

*Proof.* Since $y \mid \mathbf{x} \sim \mathcal{N}(f(\mathbf{x}), s^2(\mathbf{x}))$, the residual $\mu(\mathbf{x};\boldsymbol{\theta}) - y \sim \mathcal{N}(0, s^2(\mathbf{x}))$. By standardizing, we obtain:

$$Z = \frac{\mu(\mathbf{x};\boldsymbol{\theta}) - y}{s(\mathbf{x})} \sim \mathcal{N}(0, 1). \tag{4}$$

Thus, the squared residual can be obtained: $r = (\mu(\mathbf{x};\boldsymbol{\theta}) - y)^2 = s^2(\mathbf{x})Z^2$. Since $Z \sim \mathcal{N}(0, 1)$, we have $Z^2 \sim \chi^2(1)$, a specific case of Gamma distribution $Z^2 \sim \text{Gamma}\left(\frac{1}{2}, \frac{1}{2}\right)$. Since a Gamma random variable $W \sim \text{Gamma}(\alpha, \lambda)$ scaled by a constant $c > 0$ gives $cW \sim \text{Gamma}\left(\alpha, \frac{\lambda}{c}\right)$, then:

$$r \sim \text{Gamma}\left(\frac{1}{2}, \frac{1}{2s^2(\mathbf{x})}\right) \tag{5}$$

Showing that the mean of the Gamma distribution becomes $\frac{\alpha}{\lambda} = s^2(\mathbf{x})$, i.e., the aleatoric variance. □

We propose the Gamma likelihood to model the squared residual $r$, leading to the corresponding NLL loss to train the variance network:

$$\mathcal{L}_2(\boldsymbol{\phi}) = \sum_{n=1}^{N} \Big[ \log \Gamma\big(\alpha(\mathbf{x}_n;\boldsymbol{\phi})\big) - \alpha(\mathbf{x}_n;\boldsymbol{\phi}) \log \lambda(\mathbf{x}_n;\boldsymbol{\phi}) - \big(\alpha(\mathbf{x}_n;\boldsymbol{\phi}) - 1\big) \log r_n + \lambda(\mathbf{x}_n;\boldsymbol{\phi})\, r_n \Big]. \tag{6}$$

where $r$ are the above-mentioned residuals, and with the shape and rate parameters of the Gamma distribution, $\alpha(\mathbf{x};\boldsymbol{\phi}) > 0$ and $\lambda(\mathbf{x};\boldsymbol{\phi}) > 0$, being the outputs of the network. These parameters are obtained by MAP estimate, and their gradients with respect to $\alpha$ and $\lambda$ do not contain high-order terms, as shown below:

$$\nabla_{\alpha(\mathbf{x})}\mathcal{L}_2 = \sum_{n=1}^{N} \Big[ \psi\big(\alpha(\mathbf{x}_n)\big) - \log \lambda(\mathbf{x}_n) - \log r_n \Big], \quad \nabla_{\lambda(\mathbf{x})}\mathcal{L}_2 = \sum_{n=1}^{N} \Big[ -\frac{\alpha(\mathbf{x}_n)}{\lambda(\mathbf{x}_n)} + r_n \Big]. \tag{7}$$

The expected value of the Gamma distribution becomes the desired heteroscedastic variance:

$$\sigma_a^2(\mathbf{x};\boldsymbol{\phi}) = \frac{\alpha(\mathbf{x};\boldsymbol{\phi})}{\lambda(\mathbf{x};\boldsymbol{\phi})} \tag{8}$$

### 3.3.2 BAYESIAN NEURAL NETWORK TRAINING

Having determined the mean and aleatoric variance estimates, we then train a BNN with a warm-start for the mean, and assuming fixed aleatoric variance that is obtained from Equation (8). In principle, the same network architecture can be used for the BNN and the mean network[2]. The posterior of the BNN is determined by Bayes' rule, and it is proportional to the product of likelihood and prior:

$$p(\boldsymbol{\theta} \mid \mathcal{D}) \propto p(\mathcal{D} \mid \boldsymbol{\theta})\, p(\boldsymbol{\theta}) \tag{9}$$

where $p(\boldsymbol{\theta})$ is the prior over the neural network parameters, and $p(\mathcal{D} \mid \boldsymbol{\theta})$ is the likelihood for the observations. In this work, we consider a Gaussian prior with zero mean and unit variance, and the likelihood is given by Equation (2), i.e., it arises from a Gaussian observation distribution with heteroscedastic noise. The logarithm of the posterior becomes:

$$\log p(\boldsymbol{\theta} \mid \mathcal{D}) = \sum_{n=1}^{N} \left[ \log \frac{1}{\sqrt{2\pi\sigma_a^2(\mathbf{x}_n;\boldsymbol{\phi})}} - \frac{(\mathbf{y}_n - \mu(\mathbf{x}_n;\boldsymbol{\theta}))^2}{2\sigma_a^2(\mathbf{x}_n;\boldsymbol{\phi})} \right] + m \log \frac{1}{\sqrt{2\pi/\kappa}} - \frac{\kappa}{2}\|\boldsymbol{\theta}\|^2 \tag{10}$$

where $\kappa = 1.0$ is the precision of the prior distribution and $m$ is the length of $\boldsymbol{\theta}$. Note that Equation (10) is the same as Equation (2), but now we explicitly include the prior terms. Appendix B

---

[2]Estimating epistemic uncertainty with BNNs can require a network with wider hidden layers when compared to a deterministic network that only estimates the mean. So, choosing a smaller mean network in Step 1 is also possible. However, we like the simplicity of not introducing a new network.

summarizes common Bayesian inference strategies, including the MCMC-based methods that sample directly from Equation (10), and VI methods that minimize the evidence lower bound (ELBO) (Appendices B.1 and B.2, respectively). Importantly, since the aleatoric variance $\sigma_a^2(\mathbf{x}_n; \boldsymbol{\phi})$ is fixed from step 2 and is regarded as constant, it converges to the correct mode $\mu(\mathbf{x}_n; \boldsymbol{\theta})$, and has better and faster convergence (no gradient issue).

From the PPD of the BNN, we then estimate the mean, aleatoric, and epistemic variances (disentangled) for any unseen point $\mathbf{x}'$ based on Bayesian model averaging:

$$
\mathcal{N}\left( \mathbf{y}' \,\middle|\, \underbrace{\mathbb{E}_{p(\boldsymbol{\theta}|\mathcal{D})}\big[\mu(\mathbf{x}'; \boldsymbol{\theta})\big]}_{\text{Predictive Mean}}, \underbrace{\sigma_a^2(\mathbf{x}'; \boldsymbol{\phi})}_{\text{Aleatoric}} + \underbrace{\mathbb{V}_{p(\boldsymbol{\theta}|\mathcal{D})}\big[\mu(\mathbf{x}'; \boldsymbol{\theta})\big]}_{\text{Epistemic}} \right) \tag{11}
$$

## 4 EXPERIMENTS

**Datasets** We consider four distinct sets of datasets (a total of 18 datasets): (1) the previously discussed one-dimensional illustrative example (Skafte et al., 2019), with results presented in Appendix D.1; (2) UCI regression datasets (Hernandez-Lobato and Adams, 2015); (3) large-scale image regression datasets (Gustafsson et al., 2023); and (4) our own dataset obtained from computer simulations of materials undergoing history-dependent deformations (material plasticity law discovery dataset). This last dataset is the only one where we know the ground-truth of aleatoric uncertainty for assessing the estimation accuracy, and assumes particular importance for this investigation. Readers interested in more details about the dataset are referred to Appendix F.

**Baselines.** We compare our strategy of cooperative training – labeled as **Ours: VeBNN** – with representative methods using joint training:

1. **ME (MSE)**: standard training of a Mean Estimation network using MSE loss. **Purpose**: as this network only estimates the mean, it establishes a baseline for this estimate when compared with methods that also estimate uncertainties;

2. **MVE ($\beta$-NLL)**: MVE network with $\beta$-NLL loss (Seitzer et al., 2022). **Purpose**: assessing estimation of mean and aleatoric uncertainty, without considering epistemic uncertainty;

3. **MVE (Natural)**: MVE network with natural re-parameterized NLL loss (Immer et al., 2023). **Purpose**: same as (2) but using different loss.

4. **Evidential**: Deep Evidential regression (Amini et al., 2020). **Purpose**: comparing with a method that attempts to simultaneously estimate mean, as well as aleatoric and epistemic uncertainties;

5. **MVE (Ensembles)**: deep ensembling of jointly trained MVE networks (Lakshminarayanan et al., 2017). **Purpose**: assess ability of this inference method to simultaneously estimate mean, as well as aleatoric and epistemic uncertainties.

6. **MVE (MC-Dropout)**: jointly trained MVE network with inference by Monte Carlo Dropout (MC-Dropout) (Kendall and Gal, 2017). **Purpose**: same as (5).

7. **MVE (BBB)**: jointly trained MVE network with inference by Bayes By Backpropagation (BBB) (Blundell et al., 2015). **Purpose**: same as (5);

8. **MVE (pSGLD)**: jointly trained MVE network by inference by preconditioned Stochastic Gradient Langevin Dynamics (pSGLD) (Li et al., 2016). **Purpose**: same as (5).

Details about accuracy metrics, additional results, and training are presented in Appendix C, Appendix D, and Appendix E, respectively.

### 4.1 REAL WORLD DATASETS WITH UNKNOWN ALEATORIC UNCERTAINTY

An important challenge in assessing the performance of methods that disentangle uncertainties is that existing datasets do not provide the ground-truth aleatoric uncertainty to assess the estimation accuracy. In this section, we consider 16 different datasets that have been used in different papers on the topic, despite having **unknown aleatoric uncertainty**. Therefore, we caution that **the

**datasets of this section can only be used to assess the quality of the mean and total uncertainty estimations**.

**UCI regression datasets**   The results for the UCI regression datasets are summarized in Table 1 (and also in Table 3, Table 4, and Table 5 of Appendix D.2) according to the **Test log-likelihood (TLL)**, **Root Mean Square Error (RMSE)**, **Test coverage (TC)**, **test interval length (TIL)** metrics, respectively. VeBNN consistently improves both TLL and RMSE across all inference methods, showing a clear advantage compared to jointly training a single MVE network. Interestingly, the cooperative training strategy is capable of transforming a "weak" inference method (BBB) and achieve competitive performance when compared with other VeBNN inference methods – see differences in performance from MVE (BBB) to VeBNN (BBB). The test coverage is close to the target value of $0.95$ for nearly all methods, indicating the absence of distribution shift. Interestingly, joint training leads to lower TLL (i.e., worse) than the corresponding cooperative straining strategy across all inference methods on the *Yacht* problem. However, this improvement comes at the cost of worse mean prediction and a larger test interval length, indicating that jointly trained MVE overfits the noise.

Table 1: **TLL ($\uparrow$) results on the UCI regression datasets.** Each entry reports the mean TLL with the standard deviation in parentheses. For each dataset, the best method is marked in **bold**, and the second-best in **bold**. A superscript $*$ indicates that the best method is significantly better than the second-best (Wilcoxon test (Wilcoxon, 1945; Demšar, 2006)). Within each inference family, the best-performing variant is italicized; A superscript $+/-$ indicates significantly better or worse performance compared to the other variants in the same family ($p < 0.05$).

| Methods | Carbon (10721, 7,1) | Concrete (1030, 8,1) | Energy (768,8,2) | Boston (506,13,1) | Power (9568,4,1) | Superconduct (21263,81,1) | Wine-Red (1599, 11,1) | Yacht (308,6,1) |
|---|---|---|---|---|---|---|---|---|
| MVE ($\beta_{\mathrm{NLL}} = 1.0$) | 3.22 (0.50) | -3.11 (0.14) | -1.09 (0.29) | -2.94 (0.55) | -2.79 (0.06) | -3.80 (0.08) | **-0.95 (0.06)** | -2.18 (1.34) |
| MVE (Natural) | **3.88 (0.24)** | -3.07 (0.16) | -1.12 (0.15) | -2.74 (0.37) | -2.76 (0.04) | -3.47 (0.12) | -0.96 (0.07) | -1.69 (1.84) |
| Evidential | -4.98 (13.79) | -3.22 (0.17) | -1.62 (0.31) | -2.84 (0.34) | -2.78 (0.06) | -3.65 (0.05) | -1.14 (0.25) | -1.63 (0.76) |
| MVE (Ensembles) | -7.72 (44.03) | -3.01 (0.24) | ***-0.85 (0.23)*** | -2.71 (0.29) | -2.78 (0.07) | *-3.43 (0.06)* | -1.05 (0.06) | *-0.92 (0.79)* |
| **Ours: VeBNN (Ensembles)** | *3.87 (0.67)* | ***-2.88 (0.15)***⁺ | -0.90 (0.08) | ***-2.54 (0.32)*** | ***-2.75 (0.04)*** | -3.51 (0.04)⁻ | ***-0.95 (0.06)***⁺ | -1.13 (0.53)⁻ |
| MVE (MC-Dropout) | 2.76 (0.12) | -3.02 (0.14) | -1.69 (0.22) | -2.69 (0.37) | -2.80 (0.03) | -3.48 (0.19) | -0.96 (0.02) | ***-0.16 (0.32)**** |
| **Ours: VeBNN (MC-Dropout)** | *3.19 (0.40)*⁺ | *-2.95 (0.12)* ⁺ | *-1.33 (0.07)*⁺ | ***-2.42 (0.23)***\*⁺ | *-2.79 (0.04)*⁺ | *-3.40 (0.20)*⁺ | -0.99 (0.08)⁻ | ***-0.76 (0.32)***⁻ |
| MVE (BBB) | -0.94 (0.01) | -140.59 (57.47) | -40.19 (12.11) | -47.97 (23.90) | -68.78 (51.33) | -980.85 (896.75) | -2.58 (0.75) | -174.45 (96.93) |
| **Ours: VeBNN (BBB)** | *3.23 (0.96)*⁺ | *-3.11 (0.23)*⁺ | *-1.26 (0.32)*⁺ | *-2.67 (0.29)*⁺ | *-2.82 (0.03)*⁺ | *-3.60 (0.07)*⁺ | ***-0.95 (0.07)***⁺ | *-1.80 (0.80)*⁺ |
| MVE (pSGLD) | -2.43 (17.66) | -3.07 (0.29) | -1.20 (0.48) | -2.57 (0.26) | -2.75 (0.07) | *-3.41 (0.20)* | -0.97 (0.11) | *-1.00 (0.35)* |
| **Ours: VeBNN (pSGLD)** | ***4.04 (0.78)***⁺ | *-2.87 (0.18)*⁺ | *-0.85 (0.10)*⁺ | *-2.57 (0.21)* | ***-2.74 (0.04)*** | ***-3.41 (0.09)*** | ***-0.95 (0.06)*** | -1.27 (0.57) |

**Image regression datasets**   We considered 8 large-scale image regression datasets under real-world distribution shifts (Gustafsson et al., 2023) that evaluate the mean and total uncertainty estimations trained on ResNet 34 architecture (He et al., 2016). The TLL results are reported in Table 2, while the RMSE, TC, and TIL results are presented in Table 6, Table 7, and Table 8, respectively (Appendix D.3). Results for BBB and MC-Dropout are omitted due to convergence issues. The MVE (Ensembles) baseline is implemented with $\beta_{\mathrm{NLL}} = 0.5$, which provides the strongest competing performance for this dataset. Across the *Cells* and *Chair* datasets, both without distribution shift, VeBNN shows consistent improvements over all metrics for both Ensembles and pSGLD. More interestingly, on the remaining datasets that exhibit distribution shift, VeBNN (Ensembles) achieves higher TLL and better test coverage, demonstrating a stronger ability to deal with shifted data. We also note that VeBNN (pSGLD) performs comparably to MVE (Ensembles), despite the fact that MVE (pSGLD) tends to overfit the aleatoric variance.

Table 2: **TLL ($\uparrow$) results on image regression datasets.** Each entry reports the mean TLL, with the standard deviation in parentheses.

| Methods | Cells | Cells-Gap | Cells-Tail | Chair | Chair-Gap | Chair-Tail | Skin | Aerial |
|---|---|---|---|---|---|---|---|---|
| MVE ($\beta_{\mathrm{NLL}} = 0.5$) | -2.60 (0.31) | -5.21 (1.68) | -16.65 (10.46) | -0.50 (0.47) | -17.46 (8.27) | -79.55 (71.32) | -8.19 (0.50) | -8.02 (0.52) |
| MVE (Natural) | -7.38 (3.49) | -973.22 (2163.47) | -6.53 (0.50) | -4.70 (0.01) | **-4.69 (0.01)** | **-5.07 (0.05)** | -2345.04 (4671.44) | -8.07 (0.35) |
| Evidential | -2.68 (0.17) | -5.73 (2.05) | -20.03 (6.60) | -0.51 (0.43) | -14.25 (4.94) | -62.44 (35.44) | -8.79 (0.63) | -8.65 (0.33) |
| MVE (Ensembles) | -2.45 (0.06) | -3.89 (0.14) | -6.06 (0.78) | -0.32 (0.23) | -5.86 (2.37) | -93.40 (32.28) | -7.61 (0.12) | -7.76 (0.27) |
| VeBNN (Ensembles) | ***-2.31 (0.13)*** | ***-3.68 (0.15)*** | -5.19 (0.28) | ***-0.09 (0.24)*** | *-5.36 (3.46)* | -71.10 (33.66) | -7.59 (0.13) | ***-7.44 (0.05)***⁺ |
| MVE (pSGLD) | -3.53 (0.56) | -3.84 (0.39) | ***-4.84 (0.57)*** | -2.84 (0.22) | ***-3.45 (0.34)**** | ***-4.69 (0.38)*** | -7.42 (0.07) | -8.94 (0.54) |
| VeBNN (pSGLD) | ***-2.42 (0.11)***⁺ | ***-3.56 (0.18)*** | -4.97 (1.86) | ***-0.14 (0.16)***⁺ | -6.02 (3.19)⁻ | -46.56 (30.92)⁻ | ***-7.38 (0.14)*** | ***-7.42 (0.13)***⁺ |

In Appendix D.3, we further show that VeBNN increases epistemic uncertainty in regions affected by distribution shift, indicating the expected reduction in model confidence. However, both aleatoric

and epistemic uncertainties increase simultaneously, whereas only epistemic uncertainty is expected to rise under a distribution shift. This indicates that part of the uncertainty increase may be attributed to undesirable aleatoric uncertainty inflation. In addition, MVE (Ensembles) is highly sensitive to the choice of $\beta$. As shown in Appendix D.3, when using the original NLL loss, MVE (Ensembles) tends to overfit the data noise while underfitting the mean.

## 4.2 New dataset with known aleatoric uncertainty: material plasticity law discovery

**Compare to baselines.** The results for this dataset obtained based on the GRU architecture are shown in Figure 2.[3] Note that results using BBB inference are not available due to lack of convergence. The reader is also referred to Figure 24 in Appendix F that shows a typical ground truth response with one input sequence (material deformation path) and corresponding output sequence (stochastic material response along that path) where each path results from 100 points obtained from a Physics simulator.

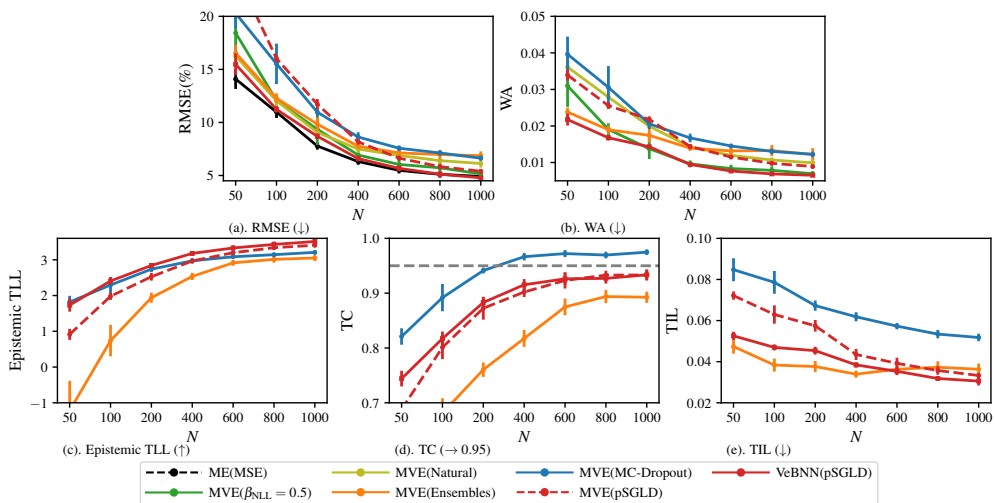

Figure 2: **Accuracy metrics obtained for the plasticity law discovery dataset considering a training set with different number $N$ of training sequences**. The Wasserstein distance (WA) represents the closeness of the estimated aleatoric uncertainty distribution to the ground truth distribution. All metrics are computed by repeating the training of each method 5 times, with random resampling of points from the training datasets.

Figure 2 shows the accuracy metrics obtained with different training sequences taken from the training dataset where $N$ increases from 50 to 1000. As expected, more training data improves accuracy for all methods, but it is clear that the proposed VeBNN (pSGLD) method achieves better predictions for all metrics, including the jointly trained MVE (pSGLD). Note that the proposed VeBNN (pSGLD) performs comparably well to the ME (MSE) when estimating the mean (RMSE metric), while reaching similar Wasserstein distance to the MVE ($\beta$-NLL $= 0.5$) network, the best method among jointly trained MVEs with MAP estimation, when estimating the aleatoric uncertainty. Importantly, the improvement of Epistemic TLL, together with the increase in test coverage is also accompanied by a decrease in the epistemic TIL. In contrast, jointly trained MVE models with Bayesian inference methods such as MVE (Ensembles), MVE (MC-Dropout), and MVE (pSGLD) exhibit lower performance in both RMSE and WA. Their overestimation of aleatoric uncertainty causes them to inflate the epistemic uncertainty (larger TIL), and results in test coverage and TLL that only become comparable to the VeBNN when the number of samples is large ($N > 600$).

We include the results of Evidential learning in a separate figure (Figure 15) due to its considerably lower performance, and we provide a detailed comparison between VeBNN (Ensembles) and

---

[3]We note that some lines are absent in some subfigures because not every method is capable of predicting the mean, aleatoric, and epistemic uncertainties simultaneously, which results in different sets of curves across the metrics.

VeBNN (MC-Dropout) with their jointly trained MVE counterparts in Appendix D.4, where similar conclusions can be reached.

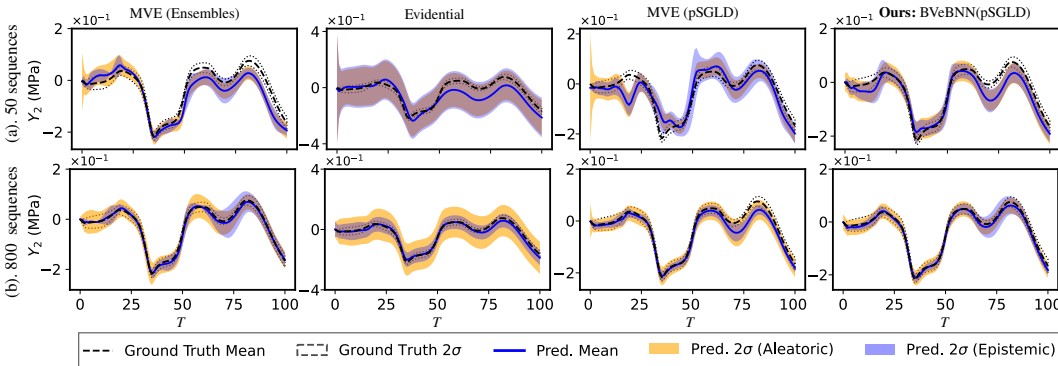

Figure 3: **Predictions of different methods on plasticity law discovery dataset.** We randomly pick one test point from the dataset and show the entire third component $\mathbf{y}_2$ of 100-time steps when considering 50 and 800 training sequences, respectively.

Figure 3 shows the predictions of the proposed VeBNN (pSGLD) and its correct identification of the disentangled data uncertainties, which improves as the training data increases (upper column to bottom column). As expected, the proposed VeBNN (pSGLD) outperforms the jointly trained MVE (pSGLD) for the same number of training sequences, and the estimated epistemic uncertainty decreases with increasing training data. Jointly trained MVE (Ensembles) perform well when considering larger training sets (800 training sequences), but performance deteriorates when compared with VeBNN (pSGLD) for 50 training sequences (this is also seen in Figure 2). On the contrary, Evidential has significant difficulties with this problem. Predictions for other methods are given in Figure 18, in which the proposed cooperative training strategy also considerably improves predictions when using MC-Dropout as inference.

We also observe in Figure 2 that the aleatoric uncertainty estimated by the VeBNN (pSGLD) converges for $N > 400$, since WA remains stable. Furthermore, the VeBNN correctly disentangles the uncertainties, as shown in Figure 3. Unfortunately, we have not found other datasets reporting the ground-truth aleatoric uncertainty like the dataset we created herein. Nevertheless, we find these results very encouraging, and we hope our dataset will motivate future developments in the field.

## 5 DISCUSSIONS

**Ablation study for Gamma Loss and number of iterations $K$.** An ablation study is conducted based on VeBNN (pSGLD) for this dataset (plasticity law discovery) because the ground-truth aleatoric uncertainty is known. The results with three performance metrics (RMSE, Epistemic TLL, and WA) are presented in Figure 4, and the remaining two are presented in Figure 21. Finding $\sigma_a^2(\mathbf{x}; \phi)$ with a Gaussian likelihood loss in Step 2 leads to worst performance due to the high-order term in the gradient calculation (Equation (3)) that can lead to $\sigma_a^2(\mathbf{x}) \to \infty$ and a degenerate local optimum for which the gradient tends to zero (Equation (3)). In contrast, the proposed Gamma NLL loss addresses this and yields a stable optimization process (Figure 4).

We also find that the proposed training strategy significantly improves upon the state-of-the-art even when training the networks only once (i.e., $K = 1$). Interestingly, we also noticed that training converged for every dataset with only one additional iteration ($K = 2$). To further explore this, Figure 19 and Figure 20 show the convergence of VeBNN along $K = 5$ iterations for plasticity law discovery datasets and *Energy* problem of UCI regression datasets, respectively. These results confirm that $K$ is merely the iteration count until convergence rather than a tunable hyperparameter.

**Size of variance estimation network** The variance estimation network used for estimating aleatoric uncertainty is trained separately from the BNN. However, we believe that this is an advantage, as the architecture required to learn aleatoric uncertainty separately is simpler than when

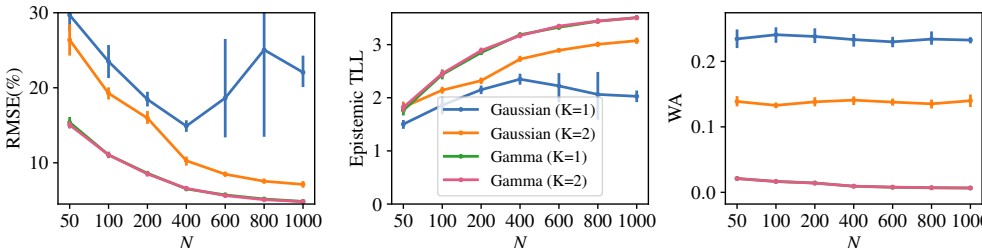

Figure 4: **Ablation study for the Gamma loss and iteration parameter $K$ using VeBNN (pS-GLD).** Curves labeled *Gamma* and *Gaussian* correspond to Step 2 training with the proposed Gamma loss (Equation (6)) and the original Gaussian NLL loss (Equation (2)), respectively, while fixing $\mu(\mathbf{x}; \boldsymbol{\theta})$ from Step 1.

training for everything at once. We considered 8 different variance network configurations and observed robust estimations of aleatoric uncertainty, as shown in Figure 22 (see Appendix D.6).

**Computational cost** As shown in Experiments, our cooperative training converges quickly: the mean network is trained once, while the variance network and Bayesian inference are performed once or twice. The mean network quickly finds a posterior mode, facilitating Bayesian training later on. Taking the inference method (pSGLD) as an example, the number of training epochs is comparable to ME and MVE training, as shown in Appendix E. In contrast, considering homoscedastic noise and treating it as a hyperparameter typically requires many full training runs and performs worse. Joined MVE training with Bayesian inference has similar efficiency, but our method consistently achieves better accuracy. Since deterministic ME and VE training is inexpensive relative to BNN training, the additional cost of our strategy is negligible and benefits from the warm start of the mean.

**Extension to multi-modal data noise.** In real-world problems, aleatoric noise may be multi-modal, making the unimodal Gaussian assumption implicit in the NLL loss (Equation (2)) insufficient. To address this, VeBNN can be seamlessly extended using Mixture Density Networks (MDNs) (Bishop, 1994) by replacing the unimodal mean network with an MDN-based mean network and adapting the variance network accordingly. This allows VeBNN to capture a broader class of predictive distributions, as illustrated in Figure 13.

**Limitations** Although the proposed method facilitates Bayesian inference for BNNs, training BNNs remains more challenging than training deterministic networks. When the posterior is non-smooth or only partially known, inference of epistemic uncertainty under fixed Gaussian prior–likelihood pairs can become unreliable. Recent advances in Imprecise Probabilistic Machine Learning (IPML), such as Credal Bayesian Deep Learning (CBDL) (Caprio et al., 2023; Wang et al., 2024) provide broader epistemic coverage but at significantly higher computational cost. Our framework may serve as a lightweight foundation that can be extended with IPML techniques in future work. Importantly, our results show that Bayesian inference is easier when aleatoric uncertainty is determined separately, as proposed herein.

## 6 CONCLUSION

We propose a novel Bayesian heteroscedastic regression strategy based on cooperative training of variance estimation and Bayesian neural networks that disentangles epistemic and aleatoric uncertainties. The proposed method is efficient, robust, and straightforward to implement because it is simpler to train each network in isolation, while ensuring complementarity in their training. As a Bayesian method, it does not require validation data. We believe the method is scalable and applicable to real-world problems that will involve active learning and Bayesian optimization, considering data-scarce and data-rich scenarios, unlike Gaussian process regression.

## DECLARATION OF GENERATIVE AI AND AI-ASSISTED TECHNOLOGIES IN THE WRITING PROCESS

The authors used CHATGPT to IMPROVE LANGUAGE AND READABILITY. After using these tools/services, the authors reviewed and edited the content as needed and assume full responsibility for the content of the publication.

## ETHICS STATEMENT

This research does not involve human participants, personally identifiable data, or sensitive social impacts. The authors are not aware of any ethical concerns associated with the methods or experiments presented.

## REPRODUCIBILITY STATEMENT

All experimental settings, hyperparameters, and architectures are described in detail in the paper and appendix. We also provide core code to replicate the illustrative example in the supplementary material to facilitate full reproducibility.

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

## A  COOPERATIVE TRAINING OF MEAN ESTIMATION (ME) AND VARIANCE ESTIMATION (VE) NETWORKS (STEP 1 AND STEP 2)

Across the 18 datasets considered in this article, we consistently found the proposed cooperative strategy (Step 1 and Step 2) to lead to improved mean and aleatoric variance estimates. Due to space limitations, Section 2.1 only provides a short explanation about the challenges involved in jointly training mean variance estimation (MVE) networks. Therefore, this appendix further elaborates on this and provides an illustrative example in Appendix A.1, as well a theoretical justification in Appendix A.2 that supports our empirical findings.

### A.1  CANONICAL EXAMPLE: JOINT TRAINING OF MVE NETWORK VS. SEQUENTIAL TRAINING OF ME AND VE NETWORKS

Figure 5 contains the results of a canonical example for one-dimensional data. As discussed in the main text, we can jointly train mean variance estimation (MVE) networks that estimates both mean and variance simultaneously, or we can train two separate networks sequentially: an ME network and a VE network.

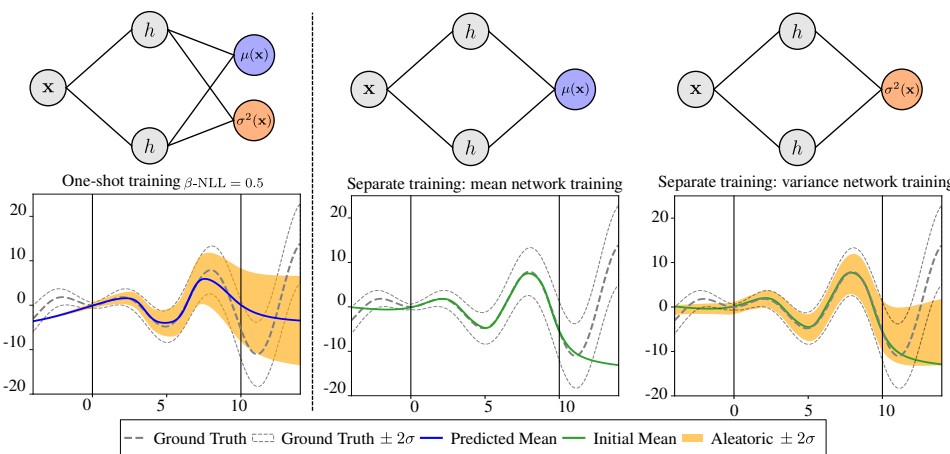

Figure 5: **Comparison between MVE network training (left figure) and separate training of ME network and variance estimation network (right figure).** The MVE network outputs $\mu(\mathbf{x})$ and $\sigma_a^2(\mathbf{x})$ and is trained by minimizing Equation (2). The ME network is trained first using Equation (2) assuming constant aleatoric uncertainty, and then the VE network is trained using Equation (6) and assuming a fixed mean obtained from the ME network.

MVE achieves worse predictions (left in Figure 5) in the region ($x > 7$) with higher aleatoric variance. As we elaborate in the next section, this occurs due to optimization challenges arising from the high-order variance term that appears in the denominator of the gradient of Equation (2). In contrast, the figure in the middle shows the result of training only the mean network (i.e., assuming constant aleatoric uncertainty), which leads to a good mean estimate – this is not surprising in light of the vast empirical evidence throughout the literature on the successes of training deterministic models that only provide a mean estimate. Furthermore, the right figure also illustrates that the subsequent training of the variance estimation network leads to an improved variance prediction that aligns well with the ground truth. This simplified example illustrates well the results obtained for all other datasets.

### A.2  THEORETICAL JUSTIFICATION OF CHALLENGES IN MVE NETWORK TRAINING

We start by restating the heteroscedastic regression problem introduced in the main section:
$$y = f(\mathbf{x}) + \epsilon(\mathbf{x}) \tag{12}$$
where $f(\mathbf{x})$ denotes the underlying noiseless ground truth function (expected mean), and $\epsilon(\mathbf{x})$ is the corresponding heteroscedastic Gaussian noise: $\epsilon(\mathbf{x}) \sim \mathcal{N}(0, s^2(\mathbf{x}))$ where $s^2(\mathbf{x})$ represents its ground truth input-dependent aleatoric variance.

The goal in this article is to learn:

1. The predictive mean: $\mu(\mathbf{x}) \approx f(\mathbf{x})$
2. Aleatoric uncertainty: $\sigma_a^2(\mathbf{x}) \approx s^2(\mathbf{x})$
3. Epistemic uncertainty: $\sigma_e^2(\mathbf{x})$ reflecting model uncertainty

### A.2.1  LOSS FUNCTIONS OF MVE, ME AND VE NETWORKS

Training a Mean-Variance Estimation (MVE) network implies joint optimization of the NLL loss written in Equation (2). We rewrite it here for completeness:

$$\mathcal{L}_{\text{MVE}}(\boldsymbol{\theta}) = \sum_{n=1}^{N} \left[ \frac{(y_n - \mu(\mathbf{x}_n; \boldsymbol{\theta}))^2}{2\sigma_a^2(\mathbf{x}_n; \boldsymbol{\phi})} + \frac{1}{2} \log\left(\sigma_a^2(\mathbf{x}_n; \boldsymbol{\phi})\right) \right] \tag{13}$$

The typical MVE architecture (Nix and Weigend, 1994) shown on the left of Figure 5 is a single network that outputs $\mu(\mathbf{x}; \boldsymbol{\theta}_{\text{joint}})$ and $\sigma_a^2(\mathbf{x}; \boldsymbol{\theta}_{\text{joint}})$, where a single set of parameters is shared $\boldsymbol{\theta} = \boldsymbol{\phi} \equiv \boldsymbol{\theta}_{\text{joint}}$.[4]

As presented in the main text, the gradient of $\mathcal{L}_{\text{MVE}}$ with respect to the mean $\mu$ is

$$\frac{\partial \mathcal{L}_{\text{MVE}}}{\partial \mu} = \sum_{n=1}^{N} \left( \frac{\mu(\mathbf{x}_n; \boldsymbol{\theta}) - y_n}{\sigma_a^2(\mathbf{x}_n; \boldsymbol{\phi})} \right) \tag{14}$$

and the gradient of $\mathcal{L}_{\text{MVE}}$ with respect to the variance $\sigma_a^2$ is

$$\frac{\partial \mathcal{L}_{\text{MVE}}}{\partial \sigma_a^2} = \sum_{n=1}^{N} \left( \frac{\sigma_a^2(\mathbf{x}_n; \boldsymbol{\phi}) - (y_n - \mu(\mathbf{x}_n; \boldsymbol{\theta}))^2}{2\left(\sigma_a^2(\mathbf{x}_n; \boldsymbol{\phi})\right)^2} \right) \tag{15}$$

This creates a coupling where the gradient with respect to the variance depends on the mean's estimation error.

Contrary to joint training, the cooperative training strategy decouples the optimization problem. As presented in Algorithm 1, training starts by considering a mean estimation (ME) network with constant variance. Therefore, the NLL loss for Step 1 is simplified because the variance does not contribute to the estimation of the mean.

- **Step 1 (Mean network training)**: The point estimate of this network is given by (omitting the regularization term):

$$\hat{\boldsymbol{\theta}} = \arg\min_{\boldsymbol{\theta}} \left[ \sum_{n=1}^{N} (y_n - \mu(\mathbf{x}_n; \boldsymbol{\theta}))^2 \right] \tag{16}$$

- **Step 2 (Variance network training)**: The VE network $\sigma_a^2(\mathbf{x}; \boldsymbol{\phi})$ is trained on the residuals $r_n = (y_n - \mu(\mathbf{x}_n; \hat{\boldsymbol{\theta}}))^2$ using the point estimate for the weights $\hat{\boldsymbol{\theta}}$ that was obtained from the ME network training in Step 1. If we use Equation (13), i.e. a Gaussian NLL, then the point estimate for the VE network parameters is (omitting the regularization term):

$$\hat{\boldsymbol{\phi}} = \arg\min_{\boldsymbol{\phi}} \left\{ \sum_{n=1}^{N} \left[ \frac{r_n}{2\sigma_a^2(\mathbf{x}_n; \boldsymbol{\phi})} + \frac{1}{2} \log\left(\sigma_a^2(\mathbf{x}_n; \boldsymbol{\phi})\right) \right] \right\} \tag{17}$$

Alternatively, as we propose in the main text, we can also use the Gamma NLL to obtain the variance estimate, as the Gamma loss does not have higher-order terms in its gradient. In that case, we minimize the loss given by Equation (6), instead of Equation (13), and estimate the variance by Equation (8).

---

[4] Alternatively, two separate networks can also be trained **simultaneously**, where one network outputs $\mu(\mathbf{x}; \boldsymbol{\theta})$ and the other $\sigma_a^2(\mathbf{x}; \boldsymbol{\phi})$, in which case each network has its set of parameters). Another option is to have a single network with a common trunk (with shared parameters) and then two heads with independent parameters that output the mean and variance estimates. Nevertheless, these architectural variations are not relevant to our analysis because in all cases there is a coupled optimization problem that needs to be solved to find the mean and variance estimates simultaneously, as discussed in this appendix.

### A.2.2 LOSS FUNCTION CONVEXITY ANALYSIS

Training neural networks involves the minimization of loss functions that are non-convex with respect to the network parameters. However, we demonstrate that the **loss of MVE networks is non-convex with respect to the mean and variance outputs (non-convexity of the last layer)**. In contrast, the ME network and the VE network (trained separately) have convex losses with respect to their outputs (mean and variance, respectively).

**Loss convexity in mean estimation network (Step 1)**: Let the network output be $u$, estimating the mean $\mu$. Then, the NLL is given by the Mean Squared Error,

$$\mathcal{L}_{ME}(u) = (y - u)^2 \tag{18}$$

and its Hessian with respect to $u$ is

$$\frac{d^2 \mathcal{L}_{ME}}{du^2} = 2 > 0 \tag{19}$$

concluding that $\mathcal{L}_{ME}(u)$ is strictly convex in $u$.

**Loss convexity in variance estimation network (Step 2)**: Let the network output be $v = \log \sigma^2$, and let the residual from the fixed mean estimate be $r = y - \hat{\mu}$. The NLL (with $\mu$ fixed) is

$$\mathcal{L}_{VE}(v) = \frac{r^2}{2} e^{-v} + \frac{1}{2} v \tag{20}$$

and its Hessian with respect to $v$ is

$$\frac{d^2 \mathcal{L}_{VE}}{dv^2} = \frac{r^2}{2} e^{-v} \geq 0 \tag{21}$$

concluding that $\mathcal{L}_{VE}(v)$ is convex in $v$ (strictly convex when $r \neq 0$).

**Loss non-convexity in MVE network**: The MVE network has a joint output loss,

$$\mathcal{L}_{\text{MVE}}(u, v) = \tfrac{1}{2}(y - u)^2 e^{-v} + \tfrac{1}{2} v \tag{22}$$

and its second derivatives with respect to $u$ and $v$ are

$$\frac{\partial^2 \mathcal{L}_{\text{MVE}}}{\partial u^2} = e^{-s}, \qquad \frac{\partial^2 \mathcal{L}}{\partial v^2} = \tfrac{1}{2}(y - u)^2 e^{-v} \tag{23}$$

and the mixed derivative is

$$\frac{\partial^2 \mathcal{L}_{\text{MVE}}}{\partial u \partial v} = \frac{\partial}{\partial v}\big(-(y - u)e^{-v}\big) = (y - u)e^{-v} \tag{24}$$

Therefore, the Hessian is

$$H = \begin{bmatrix} e^{-v} & (y - u)e^{-v} \\ (y - u)e^{-v} & \tfrac{1}{2}(y - u)^2 e^{-v} \end{bmatrix} \tag{25}$$

Its determinant is

$$\det(H) = e^{-v} \cdot \tfrac{1}{2}(y - u)^2 e^{-v} - \big((y - u)e^{-v}\big)^2 = -\tfrac{1}{2}(y - u)^2 e^{-2v} \tag{26}$$

Hence for any point with nonzero residual $(y - u) \neq 0$, $\det(H) < 0$, so $H$ is indefinite (it has eigenvalues of opposite sign). Therefore, the joint loss is *not* convex in $(u, v)$. The Hessian is only not indefinite when the exact solution $y = u$ occurs.

We conclude that training an MVE network is less stable because the loss surface has a saddle shape in the $(u, v)$ plane. Therefore, the gradients can be zero without being at a minimum, leading to worst estimates for the mean and variance, which explains the empirical observations reported in the literature (Skafte et al., 2019; Seitzer et al., 2022; Sluijterman et al., 2024) and also the results we obtain for the 18 datasets considered in this article. Interestingly, the optimization challenges are still observed when the MVE network is trained with a modified loss function that includes a gradient stopping operation, as proposed in (Seitzer et al., 2022), and summarized in Appendix A.4. In contrast, the cooperative strategy is stable and does not exhibit this behavior.

### A.3  FINITE-SAMPLE BIAS IMPROVEMENT BY COOPERATIVE TRAINING

The MVE network requires the solution of a more challenging optimization problem, as described in the previous section. This has important practical implications due to finite-sample bias.

**Theorem A.1** (Finite-Sample Bias). *In a finite-sample regime ($N < \infty$), the network will have a non-zero mean estimation bias given by $Bias(\hat{\mu}) = \mathbb{E}[\hat{\mu}(\mathbf{x})] - f^*(\mathbf{x}) \neq 0$.*

*The optimal variance estimator $\hat{\sigma}_a^2$ from the network is **a biased estimator** of the true aleatoric variance $s^2(\mathbf{x})$, as it absorbs the mean's squared bias:*

$$\mathbb{E}[\hat{\sigma}_a^2] = s^2(\mathbf{x}) + Bias(\hat{\mu}(\mathbf{x}; \hat{\boldsymbol{\theta}}))^2$$

*Proof.* The estimator $\hat{\sigma}_a^2$ is the value that minimizes the expected NLL. This occurs when the expected gradient of Equation (13) is zero, which implies:

$$\mathbb{E}[\hat{\sigma}_a^2] = \mathbb{E}\left[\left(y - \hat{\mu}(\mathbf{x}; \hat{\boldsymbol{\theta}})\right)^2\right]$$

$$= \mathbb{E}\left[\left((f^*(\mathbf{x}) - \hat{\mu}(\mathbf{x}; \hat{\boldsymbol{\theta}})) + \epsilon\right)^2\right]$$

$$= \mathbb{E}\left[(f^* - \hat{\mu})^2\right] + \mathbb{E}[\epsilon^2] + 2\mathbb{E}\left[(f^* - \hat{\mu}) \cdot \epsilon\right]$$

$$= \text{Bias}(\hat{\mu}(\mathbf{x}; \hat{\boldsymbol{\theta}}))^2 + s^2(\mathbf{x}) + 0$$

Since $N < \infty$, the bias term is non-zero. This is true whether the variance estimate results from an MVE network that was jointly trained or from a VE network that was trained cooperatively. $\square$

However, due to the convexity of the NLL loss with respect to the outputs, the cooperative (sequential) training leads to a lower bias of its mean estimator (Step 1) than the one obtained from the joint MVE estimator:

$$\text{Bias}(\hat{\mu}_{ME}(\mathbf{x}, \hat{\boldsymbol{\theta}}_{\text{ME}}))^2 < \text{Bias}(\hat{\mu}_{\text{MVE}}(\mathbf{x}, \hat{\boldsymbol{\theta}}_{\text{MVE}}))^2$$

Furthermore, we highlight that iterative training of ME and VE networks continues to reduce the bias of the mean estimator of the ME network, while improving the variance estimate. This originates from the updated variance estimate of the VE network that is used when training the ME network in the next iteration. Observe what happens after the first iteration:

- **Step 1 (Mean network training) at iteration** $k$: The point estimate of this network is updated to (omitting the regularization term):

$$\hat{\boldsymbol{\theta}}_k = \arg\min_{\boldsymbol{\theta}} \left[\sum_{n=1}^{N} \left(\frac{\mu(\mathbf{x}_n; \boldsymbol{\theta}) - y_n}{\sigma_a^2(\mathbf{x}_n; \hat{\boldsymbol{\phi}}_{k-1})}\right)\right] \tag{27}$$

  where it is important to note that the aleatoric variance $\sigma_a^2(\mathbf{x}; \hat{\boldsymbol{\phi}}_{k-1})$ is fixed from the previous iteration ($k-1$) when training the VE network, but **it is not constant** (only the log term in Equation (13) vanishes). Therefore, finding the point estimate of the ME network in this iteration $k$ remains trivial, but the mean is updated because each sample is associated to a different variance. Samples with low variance will have a large "weight" and the network will be forced to fit these points tightly; while samples with high variance will have a small weight and the network will mostly ignore them. Therefore, **the mean estimate is improved, lowering the bias of the mean estimator**.

- **Step 2 (Variance network training) at iteration** $k$: The VE network estimation will also be updated in this iteration because it is trained on the new residuals $r_{n,k} = (y_n - \hat{\mu}(\mathbf{x}_n; \hat{\boldsymbol{\theta}}_k))^2$, leading to:

$$\hat{\boldsymbol{\phi}}_k = \arg\min_{\boldsymbol{\phi}} \left\{\sum_{n=1}^{N} \left[\frac{r_{n,k}}{2\sigma_a^2(\mathbf{x}_n; \boldsymbol{\phi})} + \frac{1}{2}\log\left(\sigma_a^2(\mathbf{x}_n; \boldsymbol{\phi})\right)\right]\right\} \tag{28}$$

  Or, equivalently, updated by minimizing the Gamma loss, as previously discussed. In any case, the variance estimation improves because the mean was updated in the beginning of this iteration.

In summary, we propose the cooperative training of an ME and VE network to avoid the coupled optimization problem that imposes a non-convex loss and causes a saddle landscape in the last layer. All our experiments validate this theoretical argument. Evidently, the disadvantage of cooperative training is the introduction of an iterative procedure, but surprisingly we observe convergence after one iteration (train initial ME network with constant variance, then train VE network, and then train the ME network with the updated variance). For cases where we are interested in also estimating the epistemic uncertainty, then we do not even train the deterministic ME network again. Instead, we do Step 3 and the BNN network updates the mean and estimates the epistemic uncertainty. If a convergence test is desired, we recommend to iterate between Step 1 and Step 2, and then do the more costly Step 3 (BNN training) only once. Alternatively, it is also possible to iterate between Step 2 and Step 3 – this is a more principled iterative procedure because all estimates are being updated at each iteration, but it is also the most costly.

## A.4 MVE NETWORK WITH LOSS FUNCTION WITH GRADIENT STOPPING OPERATION

$\beta$-NLL loss (Seitzer et al., 2022) was proposed by introducing an additional variance-weighting term, which is given as follows:

$$\mathcal{L}_{\beta\text{-NLL}} = \frac{1}{N} \lfloor \sigma_a^{2\beta}(\mathbf{x}_n) \rfloor \sum_{i=n}^{N} \left[ \frac{(y_n - \mu(\mathbf{x}_n))^2}{2\sigma_a^2(\mathbf{x}_n)} + \frac{\log(\sigma_a^2(\mathbf{x}_n))}{2} \right] \quad (29)$$

where $\lfloor \cdot \rfloor$ represents the stop gradient operation. Under this setup, the gradient of the $\beta-$NLL loss becomes:

$$\nabla_\mu \mathcal{L}_{\beta\text{-NLL}} = \frac{1}{N} \sum_{n=1}^{N} \left( \frac{\mu(\mathbf{x}_n) - y_n}{\sigma_a^{2-2\beta}(\mathbf{x}_n)} \right) \quad (30)$$

$$\nabla_{\sigma_a^2} \mathcal{L}_{\beta\text{-NLL}} = \frac{1}{2N} \sum_{n=1}^{N} \left( \frac{\sigma_a^2(\mathbf{x}_n) - (y_n - \mu(\mathbf{x}_n))^2}{\sigma_a^{4-2\beta}(\mathbf{x}_n)} \right) \quad (31)$$

According to Equation (29), the variance-weighting term $\beta$ can interpolate between the original NLL loss and equivalent MSE, recovering the original NLL loss when $\beta = 0$. The gradient of Equation (30) is equivalent to MSE for $\beta = 1$.

## B  BAYESIAN NEURAL NETWORKS

In the Bayesian formalism, the posterior parameter distribution is defined as:

$$p\left(\boldsymbol{\theta} \mid \mathcal{D}\right) = \frac{p\left(\mathcal{D} \mid \boldsymbol{\theta}\right) p\left(\boldsymbol{\theta}\right)}{p\left(\mathcal{D}\right)}, \, p\left(\mathcal{D}\right) = \int p\left(\mathcal{D} \mid \boldsymbol{\theta}\right) p\left(\boldsymbol{\theta}\right) \mathrm{d}\boldsymbol{\theta} \tag{32}$$

where $p\left(\boldsymbol{\theta}\right)$ is the prior, typically a Gaussian distribution, $p\left(\mathcal{D} \mid \boldsymbol{\theta}\right)$ is the likelihood, $p\left(\mathcal{D}\right)$ is the marginal likelihood (or evidence), which integrates the likelihood over $\boldsymbol{\theta}$, and $p\left(\boldsymbol{\theta} \mid \mathcal{D}\right)$ is the posterior distribution of the parameters $\boldsymbol{\theta}$.

The BNN posterior predictive distribution (PPD) for a new data point is computed as follows:

$$p\left(y' \mid \mathbf{x}', \mathcal{D}\right) = \int p\left(y' \mid \mathbf{x}', \boldsymbol{\theta}\right) p\left(\boldsymbol{\theta} \mid \mathcal{D}\right) \mathrm{d}\boldsymbol{\theta} \tag{33}$$

where $\mathbf{x}'$ denotes the features of the unknown point, and $y'$ the corresponding target.

### B.1  MCMC SAMPLING FOR BNNS

Markov Chain Monte Carlo (MCMC) methods remain the gold standard in Bayesian inference. Usually, the prior is enforced on the weights and biases of the neural network. The most common choice is a multivariate Gaussian prior:

$$p(\boldsymbol{\theta}) = \mathcal{N}(\boldsymbol{\theta} \mid \mathbf{0}, \lambda^{-1}\mathbf{I}) \tag{34}$$

where $\lambda$ is the precision (inverse variance) of the Gaussian prior, which in deterministic networks is referred to as regularization, and $\mathbf{I}$ is the identity matrix.

The likelihood function $p\left(\mathcal{D} \mid \boldsymbol{\theta}\right)$ quantifies how well the neural network output $\mu(\mathbf{x}; \boldsymbol{\theta})$ explains the observations $\mathbf{y}$. Commonly, the observation distribution is assumed to be Gaussian:

$$p\left(\mathcal{D} \mid \boldsymbol{\theta}\right) = \prod_{n=n}^{N} \mathcal{N}(\mathbf{y}_n \mid \mu(\mathbf{x}_n; \boldsymbol{\theta}), \sigma_a^2(\mathbf{x}_n; \boldsymbol{\phi})) \tag{35}$$

where $\mu(\mathbf{x}; \boldsymbol{\theta})$ is the predictive mean of the neural network and $\sigma_a^2(\mathbf{x}; \boldsymbol{\phi})$ is the data noise variance (assumed constant when the noise is homoscedastic, or learned by another neural network parameterized by $\boldsymbol{\phi}$ as described in Section 3.3.1).

As shown in Equations (9) and (32), it is not possible to get the analytical solution because it requires integrating the posterior and marginal likelihood. However, we can obtain the following relation by omitting the denominator:

$$p\left(\boldsymbol{\theta} \mid \mathcal{D}\right) \propto p\left(\mathbf{y} \mid \boldsymbol{\theta}, \mathbf{x}\right) p\left(\boldsymbol{\theta}\right) \tag{36}$$

Substituting the prior (Equation (34)) and likelihood (Equation (35)), we obtain:

$$p\left(\boldsymbol{\theta} \mid \mathcal{D}\right) \propto \prod_{n=1}^{N} \mathcal{N}\left(\mathbf{y}_n \mid \mu(\mathbf{x}_n; \boldsymbol{\theta}), \sigma_a^2(\mathbf{x}_n; \boldsymbol{\phi})\right) \cdot \mathcal{N}(\boldsymbol{\theta} \mid \mathbf{0}, \lambda^{-1}\mathbf{I}) \tag{37}$$

$$= \prod_{n=1}^{N} \frac{1}{\sqrt{2\pi\sigma_a^2(\mathbf{x}_n; \boldsymbol{\phi})}} \exp\left(-\frac{(\mathbf{y}_n - \mu(\mathbf{x}_n; \boldsymbol{\theta}))^2}{2\sigma_a^2(\mathbf{x}_n; \boldsymbol{\phi})}\right) \cdot \frac{\lambda^{m/2}}{(2\pi)^{m/2}} \exp\left(-\frac{\lambda}{2}\|\boldsymbol{\theta}\|^2\right) \tag{38}$$

where $m$ is the number of parameters within $\boldsymbol{\theta}$. By taking the logarithmic form, we obtain

$$\log p\left(\boldsymbol{\theta} \mid \mathcal{D}\right) = \sum_{n=1}^{N}\left[\log\frac{1}{\sqrt{2\pi\sigma_a^2(\mathbf{x}_n; \boldsymbol{\phi})}} - \frac{1}{2}\frac{(\mathbf{y}_n - \mu(\mathbf{x}_n; \boldsymbol{\theta}))^2}{\sigma_a^2(\mathbf{x}_n; \boldsymbol{\phi})}\right] + m\log\frac{1}{\sqrt{2\pi/\lambda}} - \frac{\lambda}{2}\|\boldsymbol{\theta}\|^2 \tag{39}$$

where the first two terms relate to the likelihood and the final two terms only relate to the neural network parameters. In the deterministic setting, the last two often act as regularization or weight decay.

MCMC is a class of algorithms that can be used to sample from a probability distribution (Murphy, 2021). The most straightforward approach to get the posterior distribution is the random walk Metropolis-Hasting algorithm (Neal, 1995), although it suffers from high rejection rates for high-dimensional problems. To accelerate the mixing rate of the MCMC methods, the Hamiltonian Monte Carlo (HMC) was developed by Neal et al. (Neal, 1995), which leverages the concept of Hamiltonian mechanics, where the *gradient* of the target probability density function is properly utilized. Therefore, the mixing rate can be improved tremendously compared with the random walk Metropolis-Hasting algorithm. To address the scalability bottleneck of HMC, Welling et al. (Welling and Teh, 2011) developed a novel Bayesian inference approach called Stochastic Gradient Langevin Dynamics (SGLD) leveraging Stochastic Gradient Decent (SGD) (Ruder, 2017) and Langevin Monte Carlo (LMC) (Murphy, 2021).

In essence, SGLD starts with a random guess for the unknown neural network parameters $\boldsymbol{\theta}^{(0)}$ in Equation (39) and then updates the position according to the following rule (Welling and Teh, 2011):

$$\Delta\boldsymbol{\theta}_t = \frac{\eta_t}{2}\left(\nabla\log p\left(\boldsymbol{\theta}_t\right) + \nabla\frac{N}{M}\sum_{n=1}^{M}\log p(\mathcal{D}_n|\boldsymbol{\theta}_t)\right) + \boldsymbol{\zeta}_t \qquad (40)$$

where $\boldsymbol{\zeta}_t \sim \mathcal{N}(\mathbf{0}, \boldsymbol{\eta}_t)$, $\eta$ is the step size (learning rate), $N$ is the number of the total data points, $M$ is the batch size.

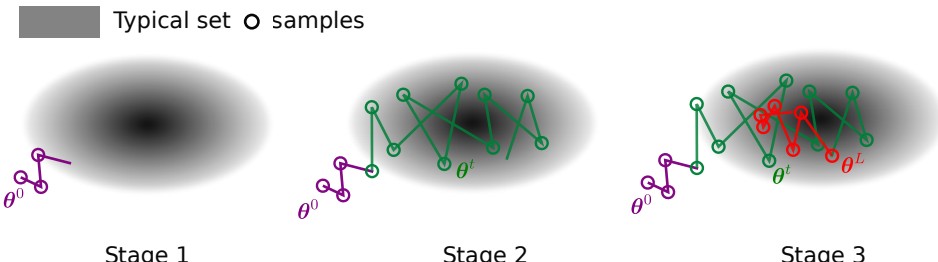

Stage 1          Stage 2          Stage 3

Figure 6: **Schematic of MCMC-based Bayesian inference.** The shaded area is the typical set, which is the region that covers the high probabilities of the posterior distribution. The samplers start with a random initialization and try to integrate the typical set. It has three stages: the first stage is to search for the typical set; the second stage is the most effective phase, exploring the typical set rapidly; and in the third stage, the sampler might converge to a single mode.

However, SGLD assumes that all parameters $\boldsymbol{\theta}$ have the same step size, which can lead to slow convergence or even divergence in cases where the components of $\boldsymbol{\theta}$ have different curvatures. A refined version of SGLD called preconditioned SGLD (pSGLD) (Li et al., 2016), was proposed to address this issue. In pSGLD (Li et al., 2016), the update rule incorporates a user-defined preconditioning matrix $G(\boldsymbol{\theta}_t)$, which adjusts the gradient updates and the noise term adaptively:

$$\Delta\boldsymbol{\theta}_t = \frac{\eta_t}{2}\left[G(\boldsymbol{\theta}_t)\left(\nabla\log p(\boldsymbol{\theta}_t) + \nabla\frac{N}{M}\sum_{n=1}^{M}p(\mathcal{D}_n|\boldsymbol{\theta}_t)\right) + \chi(\boldsymbol{\theta}_t)\right] + \boldsymbol{\zeta}_t G(\boldsymbol{\theta}_t) \qquad (41)$$

where $\chi_n = \sum_j \frac{\partial G_{n,j}(\boldsymbol{\theta})}{\partial\theta_j}$; $j = 0, \cdots, d$ describes how the preconditioner changes with respect to $\boldsymbol{\theta}$.

### B.2 VARIATIONAL INFERENCE FOR BNNS

VI is another type of Bayesian inference method that approximates the posterior distribution by a proposed distribution from a parametric family (Blundell et al., 2015; Murphy, 2021). The goal of VI is to minimize the Kullback-Leibler divergence between the true posterior distribution $p(\boldsymbol{\theta}|\mathcal{D})$ and the proposed distribution $q(\boldsymbol{\theta})$ as illustrated in Figure 7.

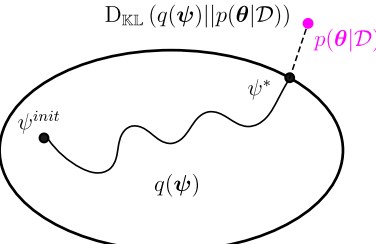

Figure 7: **Schematic of Variational Inference (Murphy, 2021).** The ellipse represents the search space of the proposed distribution, $p(\boldsymbol{\theta}|\mathcal{D})$ is the true posterior distribution, the KL divergence is the distance between the two distributions defined as $D_{\mathbb{KL}}(q(\boldsymbol{\psi})||p(\boldsymbol{\theta}|\mathcal{D}))$, and the goal is to minimize it.

Practically, $\boldsymbol{\psi}$ is regarded as the variational parameter from a parametric family $\boldsymbol{\Omega}$; therefore, the optimal $\boldsymbol{\psi}^*$ can be obtained by minimizing the KL divergence as follows (Murphy, 2021):

$$\boldsymbol{\psi}^* = \arg\min_{\boldsymbol{\psi}} D_{\mathbb{KL}}(q(\boldsymbol{\psi})||p(\boldsymbol{\theta}|\mathcal{D}))$$

$$= \arg\min_{\boldsymbol{\psi}} \mathbb{E}_{q(\boldsymbol{\psi})}[\log q(\boldsymbol{\psi}) - \log p(\boldsymbol{\theta}|\mathcal{D})]$$

$$= \arg\min_{\boldsymbol{\psi}} \mathbb{E}_{q(\boldsymbol{\psi})}\left[\log q(\boldsymbol{\psi}) - \log\left(\frac{p(\mathcal{D}|\boldsymbol{\theta})p(\boldsymbol{\theta})}{p(\mathcal{D})}\right)\right]$$

$$= \arg\min_{\boldsymbol{\psi}} \mathbb{E}_{q(\boldsymbol{\psi})}[\log q(\boldsymbol{\psi}) - \log p(\mathcal{D}|\boldsymbol{\theta}) - \log p(\boldsymbol{\theta}) + \log p(\mathcal{D})]$$

$$= \arg\min_{\boldsymbol{\psi}} \mathbb{E}_{q(\boldsymbol{\psi})}[\log q(\boldsymbol{\psi}) - \log p(\mathcal{D}|\boldsymbol{\theta}) - \log p(\boldsymbol{\theta})] + \log p(\mathcal{D}) \tag{42}$$

According to Equation (42), the optimization problem can be decomposed into two terms: the first term is the negative evidence lower bound (ELBO), and the second term is the log evidence. The log evidence is a constant term that does not depend on $\boldsymbol{\psi}$; therefore, we only need to optimize the first term:

$$\boldsymbol{\psi}^* = \arg\min_{\boldsymbol{\psi}} \mathbb{E}_{q(\boldsymbol{\psi})}[\log q(\boldsymbol{\psi}) - \log p(\mathcal{D}|\boldsymbol{\theta}) - \log p(\boldsymbol{\theta})]$$

$$= \arg\min_{\boldsymbol{\psi}} \mathbb{E}_{q(\boldsymbol{\psi})}[-\log p(\mathcal{D}|\boldsymbol{\theta})] + D_{\mathbb{KL}}(q(\boldsymbol{\psi})||p(\boldsymbol{\theta})) \tag{43}$$

Similarly as Equation (39), the first term is the negative log-likelihood of the given dataset, and the second term is the KL divergence between the proposed distribution and the prior distribution, which again can be regarded as a regularizer. The optimization problem can be solved by any optimization algorithm, such as SGD (Ruder, 2017) or Adam (Kingma and Ba, 2015). After obtaining the best $\boldsymbol{\psi}$, the variational distribution $q(\boldsymbol{\psi})$ can be used as the posterior distribution.

## C  PERFORMANCE METRICS

No single metric can thoroughly evaluate an algorithm capable of uncertainty quantification and disentanglement. Therefore, in this paper, we employ 5 metrics that aim to give a fair comparison of all selected methods. Specifically, we use Root Mean Square Error (RMSE), Test Log-Likelihood (TLL), Wasserstein distance (Kantorovich, 1960), Test Coverage (TC), and Test Interval Length (TIL), as detailed below:

- **Root Mean Square Error (RMSE)**
  - For MLP:

$$\text{RMSE} = \sqrt{\frac{1}{N_{test}} \sum_{i=1}^{N_{test}} (\bar{\boldsymbol{\mu}}_i - \mathbf{y}_i)^T (\bar{\boldsymbol{\mu}}_i - \mathbf{y}_i)} \tag{44}$$

  where $N_{test}$ is the number of test data points, $\bar{\boldsymbol{\mu}}$ is the predictive mean, and $\mathbf{y}$ is the observation values of the test points.
  - For RNN:

$$\text{RMSE} = \frac{1}{N_{test}T} \sum_{i=1}^{N_{test}} \sum_{j=1}^{T} \left( \frac{\|\bar{\boldsymbol{\mu}}_{i,j} - \bar{\mathbf{y}}_{i,j}\|_F}{\|\bar{\mathbf{y}}_{i,j}\|_F} \right) \cdot 100\%, \tag{45}$$

  where $N_{test}$ is the number of testing points, $\|\cdot\|_F$ is a Frobenius norm, $\bar{\mathbf{y}}_{i,j}$ and $\bar{\boldsymbol{\mu}}_{i,j}$ are ground truth and the predictive mean of $i^{th}$ test sample and $j^{th}$ time step, correspondingly.

- **Test Log-Likelihood (TLL)**

$$\text{TLL} = \frac{1}{N_{test}T} \sum_{i=1}^{N_{test}} \sum_{j=1}^{T} \log p(\bar{\mathbf{y}}_{i,j}|\boldsymbol{\theta}) \tag{46}$$

  We also clarify that we use the ground truth $\bar{\mathbf{y}}$ for the plasticity law dataset. Observation values $\mathbf{y}$ are used for synthetic and UCI regression datasets and $T = 1$.

- **Wasserstein distance (WA)**

$$\text{WA} = \frac{1}{N_{\text{test}}T} \sum_{i=1}^{N_{\text{test}}} \sum_{j=1}^{T} \text{W}_2 \left( p \left( \bar{\boldsymbol{\mu}}_{i,j}, \boldsymbol{\sigma}_{a_{i,j}}^2 \mathbf{I} \mid \boldsymbol{\theta}, \boldsymbol{\phi} \right), q \left( \bar{\mathbf{y}}_{i,j} \right) \right) \tag{47}$$

  where $p \left( \bar{\boldsymbol{\mu}}_{i,j}, \boldsymbol{\sigma}_{a_{i,j}}^2 \mathbf{I} \mid \boldsymbol{\theta}, \boldsymbol{\phi} \right)$ and $q \left( \bar{\mathbf{y}}_{i,j} \right)$ are the predictive and ground truth distributions, respectively. Since the Gaussian assumption is applied, we can rewrite Equation (47) into:

$$\text{WA} = \frac{1}{N_{\text{test}}T} \sum_{i=1}^{N_{\text{test}}} \sum_{j=1}^{T} \sqrt{(\bar{\mathbf{y}}_{i,j} - \bar{\boldsymbol{\mu}}_{i,j})^T (\bar{\mathbf{y}}_{i,j} - \bar{\boldsymbol{\mu}}_{i,j}) + \left( \boldsymbol{\sigma}_{i,j}^2 - \boldsymbol{\sigma}_{a_{i,j}}^2 \right)^T \left( \boldsymbol{\sigma}_{i,j}^2 - \boldsymbol{\sigma}_{a_{i,j}}^2 \right)} \tag{48}$$

- **Test Coverage (TC)**
  Given a confidence level $\alpha$, we can get the predictive confidence interval for the test dataset calibrated by the validation dataset.

$$\hat{C}_\alpha = [\mu(\mathbf{x}; \boldsymbol{\theta}) - l_\alpha \sigma(\mathbf{x}), \mu(\mathbf{x}; \boldsymbol{\theta}) + l_\alpha \sigma(\mathbf{x})], \tag{49}$$

  With the predictive confidence interval, we can obtain the test coverage with the following formula:

$$\text{Test coverage} = \frac{1}{N_{test}} \sum_{i=1}^{N_{test}} \mathbb{I}\{y_i' \in \hat{C}_\alpha(\mathbf{x}, \mathcal{D}_{test})\}. \tag{50}$$

- **Test Interval Length (TIL)** The test interval length with a given confidence interval $\alpha$ is given by

$$\hat{C}_\alpha = 2l_\alpha \sigma(\mathbf{x}), \tag{51}$$

**Uncertainty calibration**   We calibrate the predicted uncertainty using a validation dataset such that every method has the same test coverage (set to be 0.95) on the validation dataset by finding a constant $c$ to adjust the total variance (Guo et al., 2017b). Then, we report the final accuracy metrics with the calibrated total uncertainty for all compared methods. We also note that the calibration is executed for both the UCI regression and the image regression datasets.

Since the plasticity-law discovery dataset provides test datasets with ground-truth aleatoric uncertainty via 100 repeated simulations per input, it allows a direct assessment of aleatoric accuracy using the WA. Therefore, using calibration would artificially distort WA and eliminate its interpretability. In addition, we wish to preserve the intrinsic behavior of the epistemic uncertainty, where it tightens as more data are observed. Therefore, we keep the calibration factor $c = 1$ for all methods, ensuring that WA and Epistemic TLL are calculated based on ground-truth mean and predicted epistemic uncertainty reflect the raw predictive behavior of each model.

## D    ADDITIONAL EXPERIMENTS RESULTS

### D.1    ILLUSTRATIVE EXAMPLE: ONE-DIMENSIONAL DATASET

**Comparison of different inference methods**    The predictions for the one-dimensional example by our method with pSGLD inference – VeBNN (pSGLD) – are shown in Figure 1. A direct comparison with MVE ($\beta$-NLL) assuming the best value for the $\beta$ hyperparameter that we found, $\beta = 0.5$, is shown in the Appendix in Figure 5. It is clear that our strategy of separately training the mean and aleatoric variance leads to better results, and avoids the need for an extra hyperparameter ($\beta$).

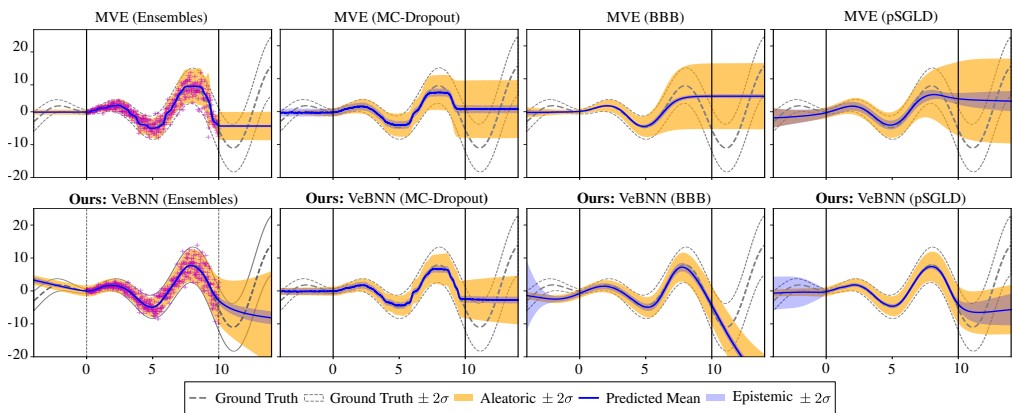

Figure 8: **Heteroscedastic regression by our method (bottom) compared to jointed trained MVE.**

Figure 8 also shows the same example when compared to other Bayesian methods with joint MVE training that are capable of predicting both uncertainties. We see a consistent improvement in the predictions with the cooperative training strategy we proposed, independently of the inference method that is chosen. Joint MVE training strategy with Bayesian inference starts to overestimate aleatoric uncertainty approximately after $x > 8$, regardless of the Bayesian inference method. This effect becomes more pronounced in the extrapolation region ($x > 10$), which reflects the difficulties that arise from the joint training process. There is a clear tendency to overestimate the aleatoric uncertainty by the joint training strategy that also affects the mean estimation. In contrast, the proposed strategy improves predictions according to all metrics. The pSGLD shows its advantage on this problem, while other inference method tends to give confident predictions even in the extrapolation regions ($x < 0$ and $x > 10$)

**Heteroscedastic noise**    We show the regression results for increasing the number of training data points from 5 to 500 based on VeBNN (pSGLD). The accuracy metrics obtained by running each case 5 times for randomly sampled training sets are shown in Figure 9, and we also visualize the fitting performance of selected methods under an arbitrary run in Figure 10.

The optimal value for the hyperparameter $\beta$ was 0.5. The experimental results reveal that MVE with $\beta = 0.5$ converges to the same mean but slower than the proposed VeBNN (pSGLD). Meanwhile, MVE (pSGLD) has difficulty in converging to either mean or aleatoric uncertainty. In contrast, the proposed strategy successfully combines the strengths of MVE and BNN to effectively model aleatoric uncertainty, mean predictions, and epistemic uncertainty. The results highlight the effectiveness of cooperative training instead of joint MVE training, especially in the case of a data-scarce scenario. Notably, the proposed method maintains a stable TLL value in the extrapolation region, outperforming alternative approaches in this critical area.

**Homoscedastic noise**    We execute a similar experiment to test homoscedastic noise cases where the data size changes from 5 to 200. Figure 11 and Figure 12 highlight similar conclusions as Appendix D.1. Again, the MVE training experiences great difficulty when the training data is scarce. MVE (pSGLD) gives really large uncertainties in the cases of $N < 30$, suggesting that it tends to give overconfident uncertainty prediction instead of converging to the mean. However, the proposed

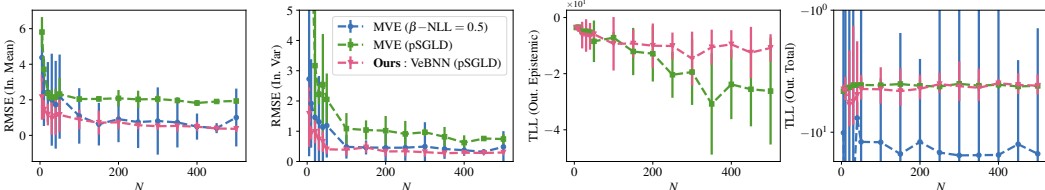

Figure 9: **RMSE and TLL convergence curves under 5 different seeds for heteroscedastic data generation.** The first figure shows the RMSE between the data values and predictive mean within the interpolation region; the second figure shows the RMSE between the noise standard deviation and the predictive one in the interpolation region since the ground truth of aleatoric uncertainty is known; and the third and fourth figures show the Epistemic TLL and Total TLL values for extrapolation test points.

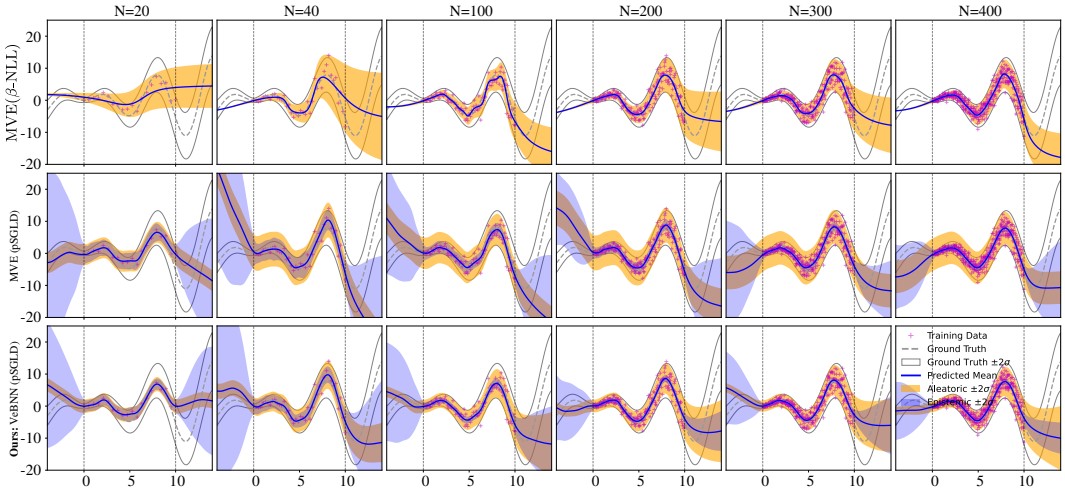

Figure 10: **Predictions of MVE ($\beta_{\text{NLL}} = 0.5$), MVE (pSGLD), and Ours: VeBNN (pSGLD) with different data points under heteroscedastic data noise.**

cooperative learning strategy has excellent performance even in the case where 5 points are used for training, which is encouraging.

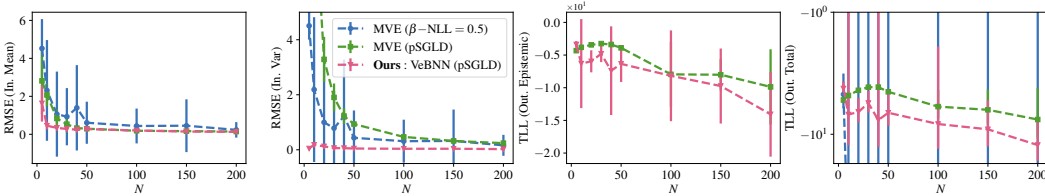

Figure 11: **RMSE and TLL convergence curves under 5 different seeds for homoscedastic data generation.**

**Extension to multi-modal aleatoric noise.** In cases where the outputs exhibit multi-modal aleatoric uncertainty (data noise), a unimodal Gaussian distribution becomes insufficient to properly capture the underlying noise structure. Therefore, in this section, we adopt the illustrative problem from (Harakeh et al., 2023) to demonstrate how the proposed VeBNN handles such problems. In short, this toy problem has two separate modes for each input $x$, and their variance change with different $x$ values. Therefore, it is different to use MVE to handle this problem with both MAP and Bayesian inference setups because they would learn a compromised mean with large aleatoric uncertainty as shown in (Harakeh et al., 2023).

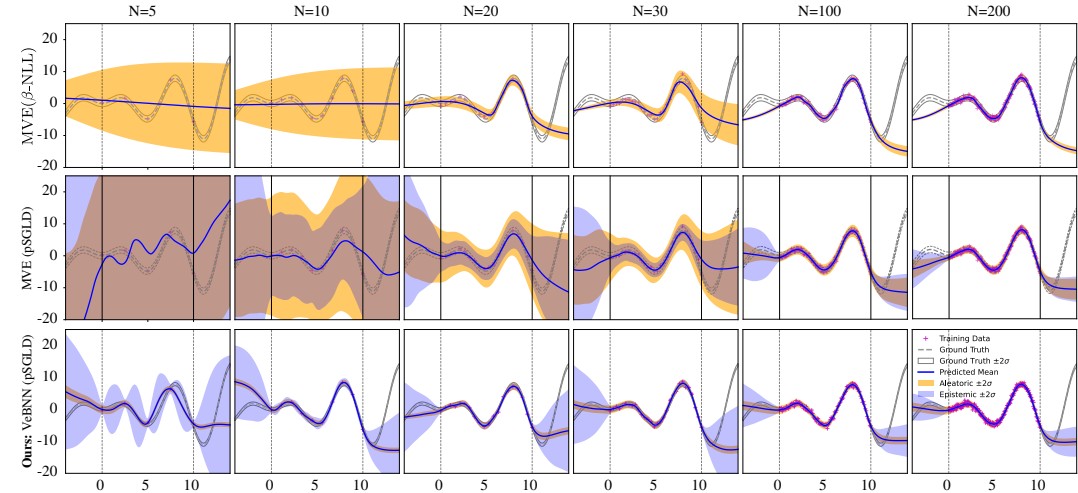

Figure 12: **Predictions of MVE ($\beta$-NLL $= 0.5$), BNN-Homo (pSGLD) and Ours: VeBNN (pSGLD) with different data points under homoscedastic data noise.**

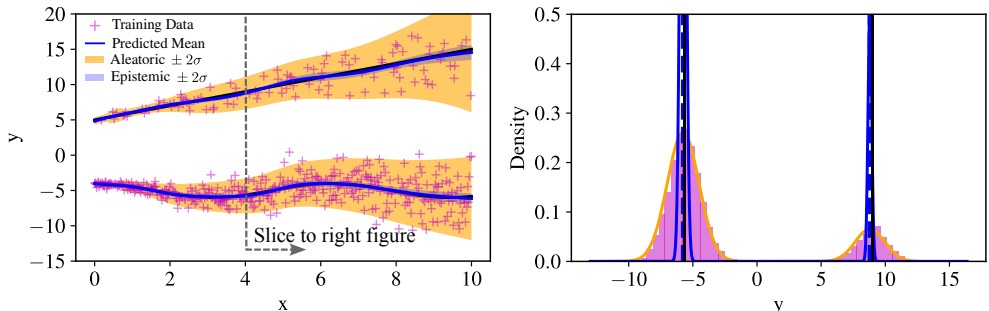

Figure 13: **Illustration of extension to multi-modal noise based on VeBNN (pSGLD) using Mixture density neural (MDN) network head.** The left figure shows the training data, and predictions with number Gaussian distribution of MDN to be 2, and the right figure shows the uncertainty identification at a test point $x = 4$.

## D.2 ADDITIONAL RESULTS FOR UCI REGRESSION DATASETS

In Section 4.1, due to article length constraints, we only presented the TLL performance where the proposed VeBNN shows clear improvements for each Bayesian inference method considered. Recall that this is demonstrated in every table by the *italic* values that highlight if cooperative training (VeBNN) or joint training (MVE) performs better for each inference method (Ensemble, MC-Dropout, BBB, and pSGLD). In this section, we summarize the results concerning RMSE, TC, and TIL in Table 3, Table 4, and Table 5, respectively.

Overall, the cooperative strategy (labeled as VeBNN) performs better for every inference method across almost all datasets. It should be mentioned that the deterministic network, ME (MSE), is competitive because Table 3 only evaluates the error in estimating the mean. However, it is still

Table 3: **RMSE ($\downarrow$) results on UCI Regression Datasets.** Each entry reports the mean RMSE, with the standard deviation shown in parentheses. For each dataset, the best method is marked in **bold**, and the second-best in **bold**. A superscript $*$ indicates that the best method is significantly better than the second-best (Wilcoxon test (Wilcoxon, 1945; Demšar, 2006)). Within each inference family, the best-performing variant is italicized; A superscript $+/-$ indicates significantly better or worse performance compared to the other variants in the same family ($p < 0.05$).

| Methods | Carbon (10721, 7,1) | Concrete (1030, 8,1) | Energy (768,8,2) | Boston (506,13,1) | Power (9568,4,1) | Superconduct (21263,81,1) | Wine-Red (1599, 11,1) | Yacht (308,6,1) |
|---|---|---|---|---|---|---|---|---|
| ME (MSE) | 0.0069 (0.0028) | 5.01 (0.79) | 0.83 (0.10) | 3.89 (0.93) | 3.88 (0.13) | **12.19 (0.44)** | **0.63 (0.05)** | 0.90 (0.36) |
| MVE ($\beta_{\text{NLL}} = 1.0$) | 0.0084 (0.0024) | 5.64 (0.73) | 0.94 (0.29) | 4.10 (1.15) | 3.91 (0.13) | 13.58 (0.39) | **0.63 (0.05)** | 2.20 (1.23) |
| MVE (Natural) | 0.0072 (0.0026) | 5.24 (0.68) | 1.06 (0.32) | 3.85 (1.48) | 3.84 (0.15) | 13.94 (5.41) | 0.64 (0.05) | 1.39 (1.58) |
| Evidential | **0.0062 (0.0028)** | 6.12 (0.79) | 2.42 (0.56) | 4.10 (1.02) | 3.91 (0.15) | 13.93 (0.52) | 0.65 (0.06) | 3.49 (2.84) |
| MVE (Ensembles) | 0.0066 (1.1927) | 5.07 (1.19) | *0.81 (0.15)* | 3.92 (0.90) | 3.87 (0.12) | *11.85 (0.48)$^*$* | 0.74 (0.09) | 1.10 (0.58) |
| **Ours:** VeBNN (Ensembles) | *0.0065 (0.0030)* | *4.83 (0.70)* | 0.83 (0.11) | *3.50 (0.96)* | *3.83 (0.14)* | 12.36 (0.40)$^-$ | *0.63 (0.05)$^+$* | *0.84 (0.36)* |
| MC-Dropout | 0.0158 (0.0017) | 5.44 (0.77) | 2.27 (0.39) | 3.94 (1.18) | 4.03 (0.12) | 13.24 (0.40) | **0.63 (0.05)** | 0.96 (0.42) |
| **Ours:** VeBNN (MC-Dropout) | *0.0092 (0.0019)$^+$* | *5.00 (0.70)$^+$* | *1.28 (0.13)$^+$* | *3.49 (1.60)* | *3.99 (0.13)* | *12.26 (0.40)$^+$* | 0.64 (0.06) | *0.79 (0.27)* |
| MVE (BBB) | 0.1748 (0.0594) | 73.61 (54.76) | 63.73 (48.89) | 530.37 (225.64) | 12.85 (8.79) | 335.87 (217.57) | 4.19 (3.72) | 572.11 (188.76) |
| **Ours:** VeBNN (BBB) | *0.0070 (0.0027)$^+$* | *5.62 (0.87)$^+$* | *1.37 (0.28)$^+$* | *3.85 (1.05)$^+$* | *4.10 (0.12)$^+$* | *13.96 (0.46)$^+$* | *0.63 (0.05)$^+$* | *1.61 (0.55)$^+$* |
| MVE (pSGLD) | 0.0066 (0.0029) | 5.76 (0.78) | 2.09 (0.32) | 3.88 (0.97) | 3.84 (0.15) | 13.63 (0.38) | 0.64 (0.06) | 1.89 (0.88) |
| **Ours:** VeBNN (pSGLD) | *0.0064 (0.0030)* | *4.70 (0.77)$^+$* | *0.77 (0.12)$^+$* | *3.64 (0.82)* | *3.79 (0.14)$^{*+}$* | *12.23 (0.45)$^+$* | **0.63 (0.05)** | *0.87 (0.35)$^+$* |

Table 4: **TC ($\rightarrow 0.95$) results on UCI regression datasets.** Each entry reports the mean, with the standard deviation in parentheses. For each dataset, the best overall performance is underlined and in bold, while the second-best is bold. Within each inference method, the best-performing variant is additionally italicized.

| Methods | Carbon (10721, 7,1) | Concrete (1030, 8,1) | Energy (768,8,2) | Boston (506,13,1) | Power (9568,4,1) | Superconduct (21263,81,1) | Wine-Red (1599, 11,1) | Yacht (308,6,1) |
|---|---|---|---|---|---|---|---|---|
| MVE ($\beta_{\text{NLL}} = 1.0$) | **0.9496 (0.0074)** | **0.9422 (0.0385)** | **0.9490 (0.0277)** | **0.9451 (0.0439)** | 0.9448 (0.0083) | 0.9508 (0.0065) | 0.9556 (0.0252) | 0.9081 (0.0757) |
| MVE (Natural) | **0.9497 (0.0063)** | 0.9379 (0.0411) | 0.9396 (0.0262) | 0.9324 (0.0553) | 0.9463 (0.0079) | 0.9506 (0.0076) | 0.9459 (0.0312) | 0.9306 (0.0680) |
| Evidential | 0.7459 (0.4419) | **0.9500 (0.0284)** | **0.9497 (0.0234)** | 0.9363 (0.0524) | 0.9508 (0.0087) | 0.9513 (0.0087) | 0.9572 (0.0238) | **0.9565 (0.0395)** |
| MVE (Ensembles) | *0.9491 (0.0069)* | 0.9291 (0.0434) | 0.9438 (0.0297) | 0.9343 (0.0437) | *0.9493 (0.0078)* | **0.9494 (0.0068)** | **0.9491 (0.0301)** | *0.9274 (0.0636)* |
| **Ours:** VeBNN (Ensembles) | 0.9478 (0.0057) | *0.9325 (0.0414)* | *0.9532 (0.0153)* | *0.9441 (0.0388)* | 0.9457 (0.0085) | 0.9513 (0.0061) | *0.9506 (0.0258)* | 0.9258 (0.0503) |
| MC-Dropout | 0.9601 (0.0160) | 0.9301 (0.0426) | 0.9373 (0.0257) | 0.9402 (0.0395) | 0.9479 (0.0089) | *0.9495 (0.0069)* | 0.9547 (0.0290) | 0.9210 (0.0568) |
| **Ours:** VeBNN (MC-Dropout) | *0.9794 (0.0073)* | *0.9354 (0.0360)* | *0.9523 (0.0196)* | *0.9431 (0.0376)* | 0.9462 (0.0084) | 0.9512 (0.0059) | 0.9569 (0.0247) | 0.9177 (0.0682) |
| MVE (BBB) | 0.9998 (0.0003) | 0.1015 (0.0473) | 0.2075 (0.0707) | 0.1598 (0.0677) | 0.1574 (0.0706) | 0.0590 (0.0219) | 0.9816 (0.0151) | 0.1048 (0.0947) |
| **Ours:** VeBNN (BBB) | *0.9474 (0.0068)* | *0.9248 (0.0499)* | *0.9477 (0.0277)* | *0.9324 (0.0578)* | *0.9477 (0.0068)* | *0.9512 (0.0061)* | *0.9528 (0.0267)* | *0.8903 (0.0853)* |
| MVE (pSGLD) | *0.9489 (0.0058)* | 0.9340 (0.0369) | 0.9412 (0.0295) | *0.9382 (0.0372)* | 0.9476 (0.0096) | 0.9492 (0.0062) | 0.9512 (0.0266) | 0.9018 (0.0664) |
| **Ours:** VeBNN (pSGLD) | 0.9473 (0.0064) | *0.9398 (0.0388)* | *0.9532 (0.0184)* | *0.9382 (0.0552)* | *0.9492 (0.0078)* | 0.9511 (0.0068) | 0.9528 (0.0284) | *0.9323 (0.0644)* |

Table 5: **TIL ($\downarrow$) results on UCI regression datasets.** Each entry reports the mean, with the standard deviation in parentheses. For each dataset, the best overall performance is underlined and in bold, while the second-best is bold. Within each inference method, the best-performing variant is additionally italicized.

| Methods | Carbon (10721, 7,1) | Concrete (1030, 8,1) | Energy (768,8,2) | Boston (506,13,1) | Power (9568,4,1) | Superconduct (21263,81,1) | Wine-Red (1599, 11,1) | Yacht (308,6,1) |
|---|---|---|---|---|---|---|---|---|
| MVE ($\beta_{\text{NLL}} = 1.0$) | 0.0366 (0.0041) | 22.22 (4.15) | 3.13 (0.55) | 18.62 (9.42) | 14.91 (0.53) | 51.27 (2.97) | **2.56 (0.28)** | 7.35 (3.56) |
| MVE (Natural) | 0.0213 (0.0016) | 20.31 (3.24) | 3.13 (0.68) | 13.07 (5.27) | 14.59 (0.54) | 39.27 (1.92) | **2.51 (0.24)** | 2.43 (1.16) |
| Evidential | 0.0292 (0.0079) | 28.61 (9.31) | 9.07 (4.12) | 17.01 (6.60) | 15.68 (0.77) | 46.55 (3.58) | 3.40 (0.86) | 13.70 (14.01) |
| MVE (Ensembles) | 0.0172 (0.0056) | 18.85 (3.35) | *2.51 (0.39)* | 13.97 (2.43) | 18.93 (0.55) | *36.91 (1.14)* | 3.05 (0.36) | 3.63 (1.50) |
| VeBNN (Ensembles) | *0.0143 (0.0009)* | *17.24 (2.98)* | 2.60 (0.28) | *12.27 (2.86)* | *14.40 (0.46)* | 38.26 (1.27) | 2.65 (0.24) | *2.32 (0.82)* |
| MC-Dropout | 0.0496 (0.0042) | *18.99 (2.61)* | 7.22 (0.79) | 14.38 (5.54) | 15.14 (0.53) | *40.11 (1.33)* | 2.62 (0.33) | 2.80 (0.96) |
| VeBNN (MC-Dropout) | *0.0304 (0.0004)* | 19.15 (2.71) | *4.13 (0.48)* | *11.43 (1.99)* | 14.86 (0.59) | 41.56 (1.59) | 2.88 (0.33) | *1.89 (0.51)* |
| MVE (BBB) | 3.93 (0.02) | *18.23 (14.56)* | 31.05 (33.95) | 249.92 (114.79) | *4.37 (1.32)* | *24.75 (11.51)* | 20.88 (18.59) | 136.95 (44.57) |
| VeBNN (BBB) | *0.0146 (0.0010)* | 19.84 (3.11) | *4.20 (1.49)* | *13.60 (4.76)* | 15.16 (0.58) | 43.83 (1.60) | 2.63 (0.27) | 4.46 (0.99) |
| End-to-end (pSGLD) | *0.0112 (0.0009)* | 20.12 (2.46) | 4.88 (0.92) | 15.22 (4.21) | *14.31 (0.44)* | 42.88 (1.61) | 2.79 (0.30) | 4.80 (2.13) |
| VeBNN (pSGLD) | 0.0141 (0.0011) | *17.71 (2.93)* | *2.45 (0.27)* | *12.60 (3.44)* | 14.66 (0.53) | *39.94 (2.00)* | 2.66 (0.26) | *2.71 (0.68)* |

slightly worse than VeBNN (pSGLD), which is the best for the UCI regression datasets. We raise awareness that the proposed cooperative training strategy consistently outperforms joint MVE training for all Bayesian inference methods that we consider (Deep Ensembles, MC-Dropout, BBB, and pSGLD). This reinforces that the proposed strategy is clearly beneficial, independent of the Bayesian inference method of choice. It is worth noting that BBB has great difficulty converging when adopting joint training for MVE, as shown in Table 3. However, by adopting the cooperative training strategy, VeBNN (BBB) reports comparable performance to state-of-the-art methods. Nevertheless, as we discuss in the main text, we recommend the use of pSGLD for the types of distributions considered herein.

## D.3 Additional results for image regression datasets

### D.3.1 Complete Results Compared to Baselines

As stated in (Gustafsson et al., 2023), the purpose of this dataset is to evaluate the uncertainty quantification capabilities of deep learning models under distribution shift. MVE (Ensembles) consistently serves as the strongest baseline across all problems, aligning with findings from (Gustafsson et al., 2023). *Cells* and *ChairAngle* are the two baseline tasks without any distribution shift between training and test sets. In these cases, the VeBNNs achieve the targeted test coverage of 95% (Table 7), while performing better by reaching larger TLL (Table 2), smaller RMSE (Table 6), and test interval length (Table 8) for both inference methods: Ensembles and pSGLD. This is relevant because independently of the inference method that is chosen, the benefits of cooperative training are noteworthy for all accuracy metrics, and training is more robust (less sensitive to the hyperparameters).

For tasks with distribution shift (Cells-Gap, Cells-Tail, ChairAngle-Gap, ChairAngle-Tail, Skin and Aerial), the VeBNN (Ensembles) reach a test coverage closer to the 95% target than other methods, while achieving better TLL. There is a slight increase in RMSE and Test interval length. This is expected; the model is tested on a region without nearby training points, therefore, its mean predictions are less trustworthy, and there is an increase in predicted epistemic uncertainty. Therefore, this would be advantageous in decision-making scenarios.

To further demonstrate this effect, we plot the predicted aleatoric and epistemic uncertainties of jointly trained MVE and VeBNN with both Ensembles and pSGLD inference in Figure 14. Although these datasets cannot be used to evaluate the quality of aleatoric uncertainty estimation, we noticed that the proposed VeBNN methods tend to have higher epistemic uncertainty in out-of-distribution regions (datasets ending with "-TAIL"), while the predicted aleatoric uncertainty remains stable even in these regions, regardless of whether Ensembles or pSGLD are used. This suggests that VeBNN may disentangle aleatoric uncertainty more effectively than existing approaches. In contrast, for both MVE (Ensembles) and MVE (pSGLD), the two uncertainty types behave similarly: they remain small within the in-distribution region and increase simultaneously when entering out-of-distribution regions.

Table 6: **RMSE ($\downarrow$) results on image regression datasets.** Each entry reports the mean RMSE, with the standard deviation shown in parentheses. A superscript $*$ indicates that the best method is significantly better than the second-best (Wilcoxon test (Wilcoxon, 1945; Demšar, 2006)). Within each inference family, the best-performing variant is italicized; A superscript $+/-$ indicates significantly better or worse performance compared to the other variants in the same family ($p < 0.05$).

| Methods | Cells | Cells-Gap | Cells-Tail | ChairAngle | ChairAngle-Gap | ChairAngle-Tail | Skin | Aerial |
|---|---|---|---|---|---|---|---|---|
| MVE ($\beta_{NLL} = 0.5$) | 5.37 (2.24) | 9.79 (2.03) | 24.97 (4.79) | 0.44 (0.22) | **2.70 (0.47)** | 6.56 (0.33) | 497.01 (49.91) | 534.29 (48.11) |
| MVE (Natural) | 1111.10 (2302.96) | 278836.40 (623369.05) | 58.32 (1.52) | 25.97 (0.18) | 26.04 (0.16) | 25.85 (0.36) | 5667.62 (8615.82) | 1011.35 (529.46) |
| Evidential | 5.56 (1.44) | **7.29 (1.63)** | 29.11 (2.79) | 0.45 (0.22) | 3.46 (0.46) | **5.68 (0.26)**$^*$ | 485.41 (23.23) | 565.96 (29.64) |
| MVE (Ensembles) | 4.82 (0.46) | *7.88 (0.84)* | **24.28 (1.55)** | 0.34 (0.08) | *2.37 (0.19)* | *6.46 (0.13)* | *446.29 (9.67)* | *456.06 (22.18)* |
| VeBNN (Ensembles) | *3.67 (0.52)* | 8.22 (0.23) | *22.96 (1.63)* | *0.24 (0.07)* | 5.53 (2.60)$^-$ | 6.66 (0.05)$^-$ | 454.71 (6.55) | 572.47 (18.85)$^-$ |
| MVE (pSGLD) | 30.74 (34.90) | 21.91 (6.92) | 414.41 (698.37) | 35.71 (31.92) | 34.45 (30.23) | 35.90 (20.74) | 463.60 (10.60) | 1003.62 (139.64) |
| VeBNN (pSGLD) | *4.44 (1.01)*$^+$ | *8.06 (0.46)*$^+$ | *39.03 (26.79)*$^+$ | *0.27 (0.08)*$^+$ | *3.88 (0.61)*$^+$ | *8.98 (1.33)*$^+$ | *452.89 (23.66)* | *479.29 (46.35)*$^+$ |

Table 7: **TC ($\rightarrow 0.95$) results on image regression datasets.** Each entry reports the mean coverage, with the standard deviation in parentheses. For each dataset, the best overall performance is underlined and in bold, while the second-best is bold. Within each inference method, the best-performing variant is additionally italicized.

| Methods | Cells | Cells-Gap | Cells-Tail | ChairAngle | ChairAngle-Gap | ChairAngle-Tail | Skin | Aerial |
|---|---|---|---|---|---|---|---|---|
| MVE ($\beta_{NLL} = 0.5$) | 0.9479 (0.0028) | 0.7224 (0.1039) | 0.5769 (0.0447) | 0.9540 (0.0026) | 0.7019 (0.0146) | 0.6638 (0.0215) | 0.8344 (0.0233) | 0.8472 (0.0426) |
| MVE (Natural) | 0.9128 (0.0845) | **0.9749 (0.0039)** | 0.5436 (0.0748) | 0.9529 (0.0020) | **0.9679 (0.0006)** | 0.6812 (0.0279) | 0.5413 (0.3497) | **0.9325 (0.1146)** |
| Evidential | **0.9502 (0.0022)** | 0.6939 (0.0482) | 0.5581 (0.0180) | 0.9815 (0.0126) | 0.7168 (0.0151) | 0.6837 (0.0081) | 0.8381 (0.0144) | 0.7694 (0.0307) |
| MVE (Ensembles) | 0.9442 (0.0056) | 0.7452 (0.0385) | 0.5604 (0.0378) | 0.9518 (0.0033) | 0.7494 (0.0160) | 0.6471 (0.0075) | 0.8371 (0.0125) | 0.8550 (0.0280) |
| VeBNN (Ensembles) | *0.9487 (0.0019)* | *0.8198 (0.0351)* | *0.5869 (0.0207)* | *0.9518 (0.0016)* | *0.7971 (0.0370)* | *0.6507 (0.0067)* | *0.8612 (0.0038)* | *0.9019 (0.0146)* |
| MVE (pSGLD) | 0.9450 (0.0048) | *0.9114 (0.0485)* | *0.8492 (0.1065)* | 0.9491 (0.0036) | *0.9278 (0.0303)* | *0.7797 (0.0536)* | *0.8625 (0.0116)* | 0.6724 (0.0600) |
| VeBNN (pSGLD) | *0.9508 (0.0048)* | 0.8416 (0.0755) | **0.7827 (0.1695)** | *0.9503 (0.0059)* | 0.7654 (0.0255) | **0.6938 (0.0325)** | 0.8561 (0.0164) | *0.9024 (0.0243)* |

Table 8: **TIL (↓) results on image regression datasets.** Each entry reports the mean interval length, with the standard deviation in parentheses. For each dataset, the best overall performance is underlined and in bold, while the second-best is bold. Within each inference method, the best-performing variant is additionally italicized.

| Methods | Cells | Cells-Gap | Cells-Tail | ChairAngle | ChairAngle-Gap | ChairAngle-Tail | Skin | Aerial |
|---|---|---|---|---|---|---|---|---|
| MVE ($\beta_{\text{NLL}} = 0.5$) | 14.65 (6.29) | 18.18 (7.43) | **18.23 (4.68)** | 1.57 (0.55) | **1.83 (0.36)** | 2.33 (0.93) | 886.30 (163.04) | 1546.45 (216.47) |
| MVE (Natural) | 196.27 (7.23) | 210.25 (21.55) | 108.23 (14.06) | 85.97 (1.58) | 89.09 (1.88) | 60.05 (2.65) | 1886.45 (1281.29) | 2539.68 (238.92) |
| Evidential | 13.05 (3.39) | **11.49 (3.25)** | **14.84 (2.72)** | 1.88 (0.66) | 2.58 (0.51) | 2.06 (0.53) | **748.02 (40.17)** | **1030.07 (31.59)** |
| MVE (Ensembles) | *12.48 (1.19)* | *16.74 (1.98)* | 22.02 (2.22) | **1.20 (0.12)** | *2.46 (0.30)* | *1.39 (0.17)* | 727.95 (27.48) | *1486.05 (62.13)* |
| VeBNN (Ensembles) | 13.52 (1.59) | 25.57 (3.32) | *19.01 (4.33)* | *1.05 (0.16)* | 10.08 (6.27) | 1.52 (0.42) | 933.98 (45.17) | 1568.95 (175.70) |
| MVE (pSGLD) | 45.73 (24.38) | 49.59 (20.10) | 439.95 (691.24) | 40.80 (15.63) | 42.06 (16.22) | 47.65 (20.68) | 917.17 (66.14) | 1562.00 (92.58) |
| VeBNN (pSGLD) | *11.87 (1.33)* | 21.58 (5.70) | *66.25 (53.73)* | *1.31 (0.19)* | *5.17 (1.52)* | *2.97 (1.30)* | *885.89 (68.61)* | *1510.50 (97.49)* |

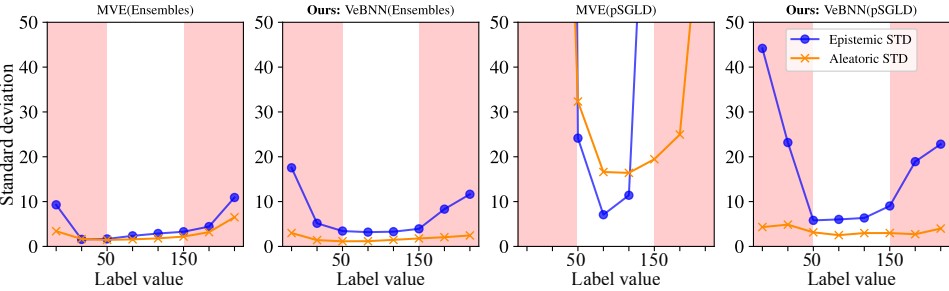

Figure 14: **Uncertainty predictions and decompositions for the *Cells-Tail* problem.** The white area represents the in-distribution labels, and the light pink area represents out-of-distribution labels.

### D.3.2 RESULTS OF MVE (ENSEMBLES) WITH ORIGINAL NLL LOSS

Although MVE (Ensembles) is the strongest baseline on the image regression datasets, we find that it is highly sensitive to the choice of $\beta$. We present the results of MVE (Ensembles) trained with the original NLL loss and compare them to those obtained with the $\beta$-NLL formulation in Table 9. As shown, the performance degrades substantially when using the original NLL loss, due to its tendency to overfit the data noise while underfitting the mean.

### D.4 PLASTICITY LAWS DATASETS

**Performance of Evidential method**   Figure 15 gives the performance metrics of the Evidential method for the plasticity laws datasets. It is observed that Evidential can only predict a comparable mean, while its aleatoric and epistemic uncertainties predictions are far worse than the methods reported in Figure 2. It starts by overfitting the aleatoric uncertainty, leading to large WA values and an inflated visualization in Figure 3. Then, the corresponding epistemic uncertainty prediction leads to an undesirable TC close to 1 as shown in Figure 15d).

**Comparison between MVE (Ensembles) and VeBNN (Ensembles).**   The comparison results are presented in Figure 16. It is clear that VeBNN (Ensembles) achieves substantially better mean and aleatoric uncertainty predictions, as evidenced by the clear performance gap in RMSE and WA. However, in this case, VeBNN (Ensembles) produces overconfident epistemic uncertainty estimates. This occurs because Deep Ensembles is not a principled Bayesian inference method; its epistemic uncertainty primarily arises from independent restarts of the training procedure. When the loss landscape is well-behaved, like step 3 of VeBNN, the ensemble members are likely to converge to very similar solutions across different runs, which significantly reduces the variability among models and therefore leads to underestimated epistemic uncertainty. Therefore, one should be cautious when using Deep Ensembles as the inference method, though it may contain enough variability in large language models due to the enormous parameter space.

**Comparison between MVE (MC-Dropout) and VeBNN (MC-Dropout).**   As it shows in Figure 17, the observation is broadly the same: VeBNN (MC-Dropout) can improve both the mean and aleatoric uncertainty. As for the epistemic uncertainty, the VeBNN (MC-Dropout) gives a small TIL,

Table 9: **Comparison of the performance of MVE (Ensembles) trained with $\beta_{\text{NLL}} = 0.5$ and Original NLL across image regression datasets.** Each entry reports the mean interval length, with the standard deviation in parentheses.

| Problem | Method | TLL($\uparrow$) | RMSE ($\downarrow$) | TC ($\rightarrow$ 0.95) | TIL ($\downarrow$) |
|---|---|---|---|---|---|
| Cells | $\beta_{\text{NLL}} = 0.5$ | -2.45 (0.06) | 4.82 (0.46) | 0.9442 (0.0056) | 12.48 (1.19) |
| | Original NLL | -2.05 (0.09) | 18.23 (8.39) | 0.9473 (0.0058) | 12.62 (1.81) |
| Cells-Gap | $\beta_{\text{NLL}} = 0.5$ | -3.89 (0.14) | 7.88 (0.84) | 0.7452 (0.0385) | 16.74 (1.98) |
| | Original NLL | -5.24 (1.24) | 8.20 (0.96) | 0.6843 (0.1021) | 14.43 (6.69) |
| Cells-Tail | $\beta_{\text{NLL}} = 0.5$ | -6.06 (0.78) | 24.28 (1.55) | 0.5604 (0.0378) | 22.02 (2.22) |
| | Original NLL | -10.23 (3.44) | 23.04 (2.00) | 0.5399 (0.0115) | 14.53 (2.33) |
| ChairAngle | $\beta_{\text{NLL}} = 0.5$ | -0.32 (0.23) | 0.34 (0.08) | 0.9518 (0.0033) | 1.20 (0.12) |
| | Original NLL | -3.80 (0.21) | 11.25 (1.70) | 0.9556 (0.0012) | 47.85 (11.68) |
| ChairAngle-Gap | $\beta_{\text{NLL}} = 0.5$ | -5.86 (2.37) | 2.37 (0.19) | 0.7494 (0.0160) | 2.46 (0.30) |
| | Original NLL | -3.69 (0.19) | 10.30 (1.57) | 0.9522 (0.0190) | 40.16 (10.35) |
| ChairAngle-Tail | $\beta_{\text{NLL}} = 0.5$ | -93.40 (32.28) | 6.46 (0.13) | 0.6471 (0.0075) | 1.39 (0.17) |
| | Original NLL | -5.08 (0.40) | 34.50 (19.52) | 0.7456 (0.0357) | 64.21 (36.42) |
| Skin | $\beta_{\text{NLL}} = 0.5$ | -7.61 (0.12) | 446.29 (9.67) | 0.8371 (0.0125) | 727.95 (27.48) |
| | Original NLL | -7.56 (0.24) | 479.44 (43.51) | 0.8734 (0.0136) | 1013.12 (133.83) |
| Aerial | $\beta_{\text{NLL}} = 0.5$ | -7.76 (0.27) | 456.06 (22.18) | 0.8550 (0.0280) | 1486.05 (62.13) |
| | Original NLL | -7.76 (0.27) | 726.16 (148.76) | 0.9473 (0.0575) | 2374.48 (184.77) |

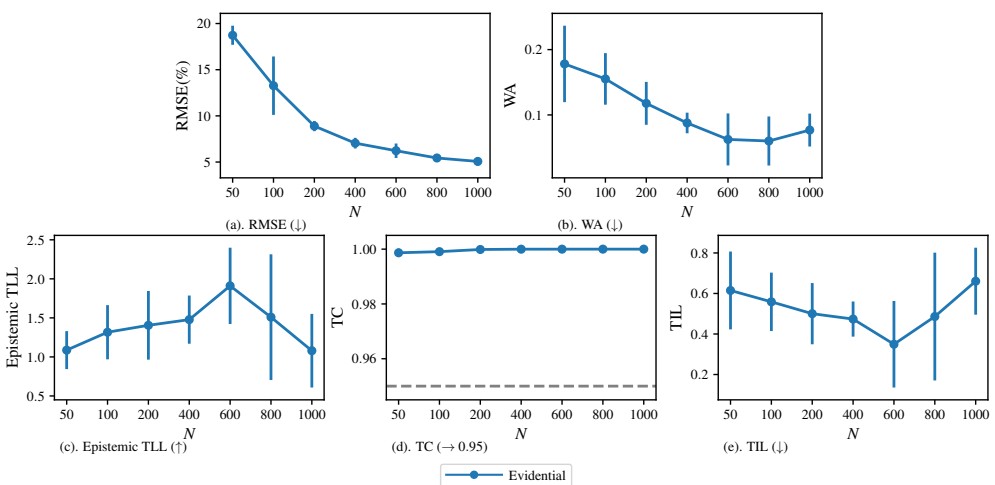

Figure 15: **Accuracy metrics obtained for the plasticity law discovery dataset considering a training set with different number $N$ of training sequences with Evidential method.**

which leads to slightly smaller Epistemic TLL and TC. With more samples, VeBNN (MC-Dropout) gets better performance as expected.

**Uncertainty identification**  Given the limited space in the main text, we present the predictions of MVE ($\beta$-NLL), MVE (MC-Dropout), and VeBNN (MC-Dropout) on the plasticity law discovery dataset in Figure 18. We first observe that MVE ($\beta$-NLL) provides reliable aleatoric uncertainty estimates when sufficient training data is available. However, its predictions—both in terms of the mean and aleatoric uncertainty—deteriorate when the training size is reduced to $N = 50$. For MVE (MC-Dropout), the predictions remain suboptimal, even with $N = 800$ training sequences. In contrast, the proposed VeBNN (MC-Dropout) improves prediction quality, particularly in estimating aleatoric uncertainty.

It is important to note that predictions for plasticity laws are expected to be smooth. The inherent bumpiness of MC-Dropout-based approaches poses challenges for deployment in Finite Element Analysis in real-world applications.

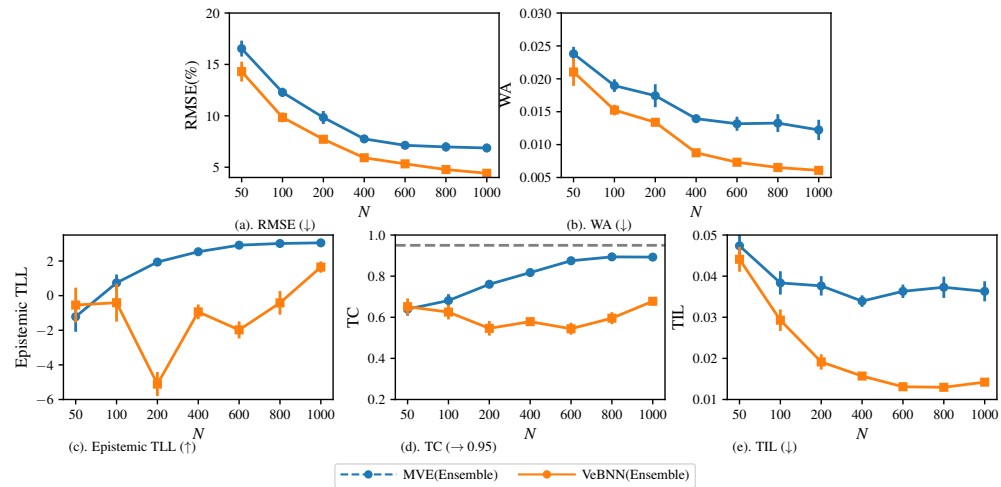

Figure 16: **Comparison between MVE (Ensembles) and VeBNN (Ensembles) for the plasticity law discovery dataset considering a training set with different number** $N$ **of training sequences**.

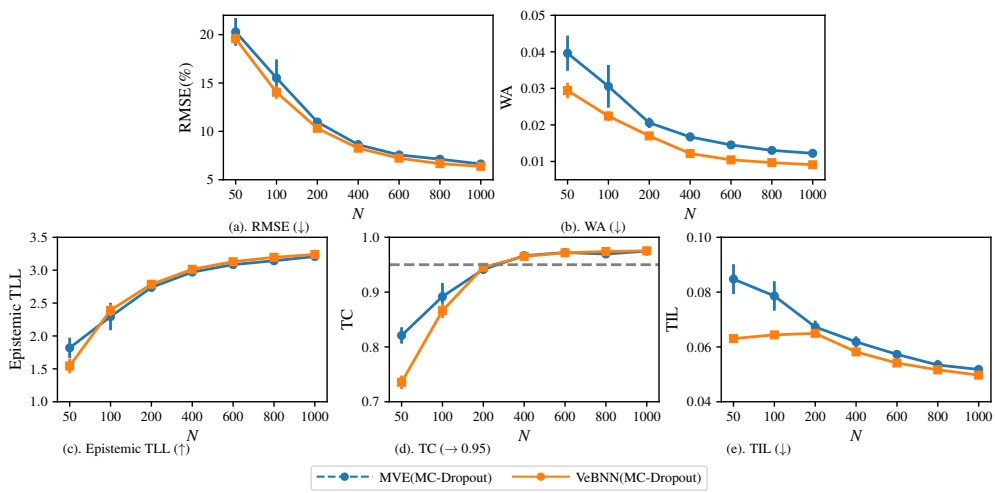

Figure 17: **Comparison between MVE (MC-Dropout) and VeBNN (MC-Dropout) for the plasticity law discovery dataset considering a training set with different number** $N$ **of training sequences**.

### D.5    ABLATION STUDY ON ITERATION $K$

We claim in Section 5 that the iteration $K$ is not a hyperparameter but a parameter, where we present an experiment conducted based on the plasticity law discovery dataset with 800 training sequences. The results are shown in Figure 19.

According to Figure 19, the proposed method exhibits rapid convergence, typically requiring only a single iteration to achieve stable performance, highlighting the effectiveness of the cooperative training strategy. Specifically, the initial warm-up of the mean network provides a favorable starting point for both **Step 2** (aleatoric modeling) and **Step 3** (Bayesian training), facilitating accurate uncertainty decomposition in subsequent stages. However, as observed in the case of seed 1, an unfavorable initialization may lead the warm-up stage to overfit noise. In such cases, additional iterations can effectively mitigate the issue and restore model performance. Based on empirical evidence, convergence occurs within 2 iterations.

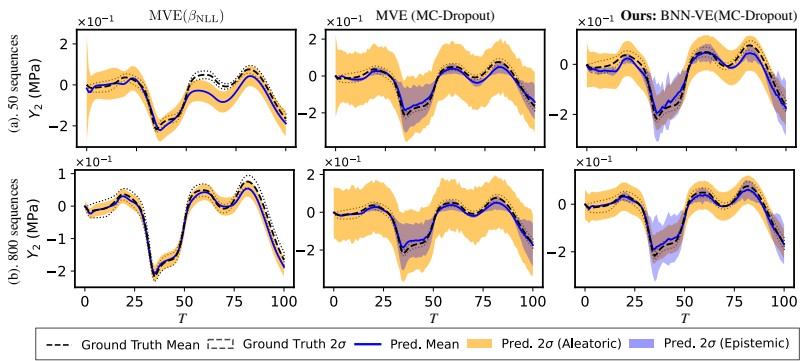

Figure 18: **Predictions of MVE ($\beta$-NLL), MVE (MC-Dropout), and VeBNN (MC-Dropout) methods on plasticity law discovery dataset.** The upper and bottom rows correspond to 50 and 800 training sequences, respectively.

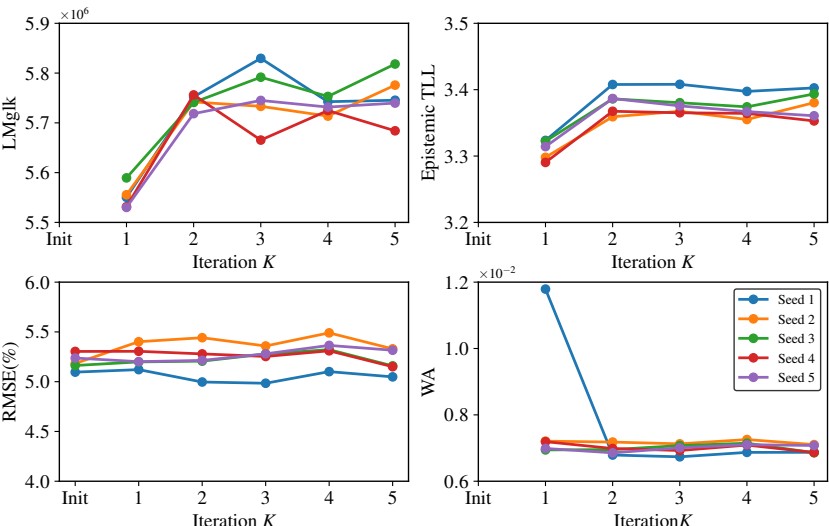

Figure 19: **Trajectory of performance metrics with respect to the iteration $K$ for 800 training sequences in the plasticity law discovery problem.** In the figures, "Init" represents the initialization of training the mean network only as described in Section 3.2, and different curves are realizations under different seeds to restart the procedure, as well as using new samples of training sequences in the dataset.

In addition, we randomly select the *Energy* problem from the UCI regression datasets and track its validation NLL over 5 iterations. The results based on VeBNN (pSGLD) are shown in Figure 20. We observe that VeBNN converges to a similar validation NLL regardless of whether the mean network in **Step 1** is properly trained. However, starting from an untrained mean network requires more iterations ($K$) to reach convergence.

**Additional figure of test coverage and interval length for Figure 4.** It further enhances the robustness of VeBNN (pSGLD) with the additional test coverage and test internal length being almost the same for different $K$ values.

D.6 ABLATION STUDY ON THE SIZE OF VARIANCE ESTIMATION NETWORK

In Section 5, we also comment on the impact of the size of the variance network, and we provide detailed results of varying the variance of the architecture configurations in Figure 22. As observed, the predictions do not change significantly when choosing different network architectures.

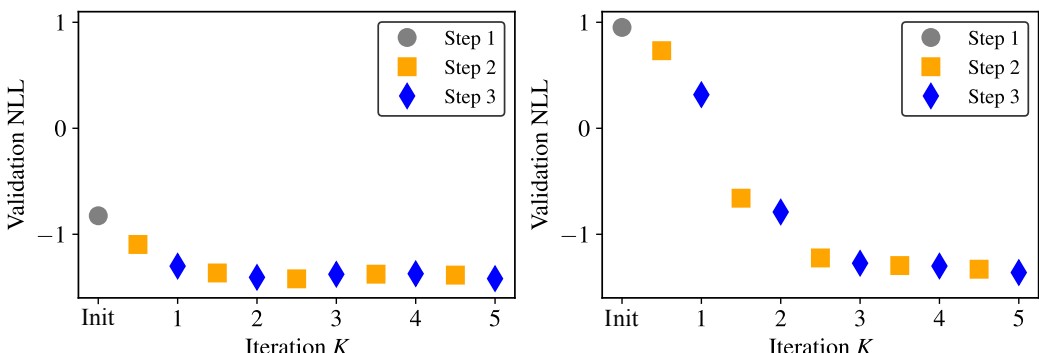

Figure 20: **Trajectory of validation loss with respect to the iteration** $K$ **for** *Energy* **problem randomly picked from UCI regression dataset.** For each iteration, the validation NLL of each step is plotted using different colors. The left subfigure corresponds to a properly trained mean network of step 1, whereas the right subfigure shows the case where the mean network of step 1 is not trained.

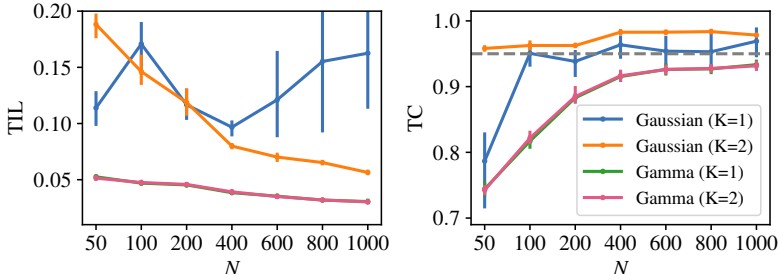

Figure 21: **TC and TIL metric performance for the ablation study for the Gamma loss and iteration parameter** $K$ **using VeBNN (pSGLD).** Curves labeled *Gamma* and *Gaussian* correspond to Step 2 training with the proposed Gamma loss (Equation (6)) and the original Gaussian NLL loss (Equation (2)), respectively, while fixing $\mu(\mathbf{x}; \boldsymbol{\theta})$ from Step 1.

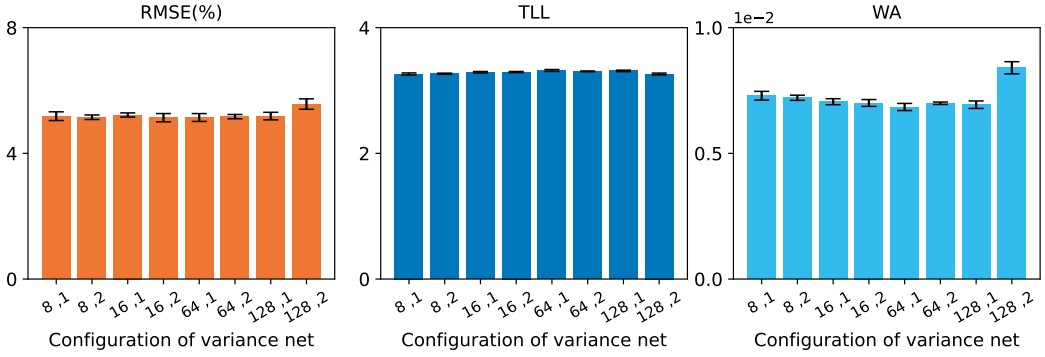

Figure 22: Barplot of all performance metrics with different variance network configurations when considering 800 training points for plasticity law discovery problem.

# E HYPERPARAMETER SETTINGS

## E.1 ILLUSTRATIVE DATASETS

**Heteroscedastic noise** In Appendix D.1, we consider the following one-dimensional example (Skafte et al., 2019):

$$y = x \sin x + 0.3 \cdot x \cdot \varepsilon_1 + 0.3 \cdot \varepsilon_2 \tag{52}$$

where $\varepsilon_1, \varepsilon_2 \sim \mathcal{N}(0,1)$. In the experiments of Appendix D.1 and Appendix D.1, we sample points uniformly from $[0, 10]$ for training. We generate 1000 points in $[0, 10]$ and 1000 points $[-4, 0] \cup [10, 14]$ to test the performance of different methods.

We depict the results of the illustrative example in Figure 1, Figure 8, and Figure 5, for which the hyperparameters are summarized as follows.

- **Architectures:** We use a two-layer multi-layer perception (MLP) with 256 neurons for the methods that only require one neural network, except BNN (BBB), where a one-layer with 50 neurons is adopted [5]. *Tanh* function is used as the activation function. We note that only the mean is outputted for the ME network, but there are two outputs, mean and aleatoric variance, for the MVE network. For the proposed cooperative learning strategy, we employ the same architecture for the BNN as the ME network. An additional one-layer MLP with 5 neurons is employed as the variance neural network to learn data uncertainty.

- **Optimizer/Inference:** We use *Adam* optimizer with a learning rate of 0.001 for 20000 epochs to optimize Equation (2) for all deterministic methods and MC-Dropout (Dropout rate is 0.1 for each layer). *Adam* is also used to optimize the ELBO of BBB for 10000 epochs with a learning rate of 0.01. As for pSGLD, we set the burn-in epoch to be 10000 and collect 100 posterior samples every 100 epochs. We also optimize the variance network for 5000 epochs with a learning rate of 0.001 and early stopping of 100 epochs for the proposed cooperative learning strategy.

- **Hyperparameter selection:** We select 70% data points for training, and the remaining data is used for finding the best hyperparameters, namely number of training epochs and $\beta$ in the case based on the NLL loss value. After identifying the best number of epochs, we use all data points to re-train the model under the best hyperparameters. For the proposed cooperative learning strategy, we feed all the data to Algorithm 1 and set the iteration $K = 5$.

**Homoscedastic noise** The homoscedastic noise case has the same ground truth, but instead of input-dependent noise, we consider constant noise, expressed as:

$$y = x \sin x + 0.5 \cdot \varepsilon_2 \tag{53}$$

where $\varepsilon_2 \sim \mathcal{N}(0,1)$.

## E.2 UCI REGRESSION DATASETS

We carefully reviewed the experiments and hyperparameters that were found in other studies using UCI regression datasets (Skafte et al., 2019; Immer et al., 2023; Seitzer et al., 2022). We used similar configurations to these studies. Our dataset splits and randomizations can be found in our code for reproducibility because the UCI dataset results can be sensitive to data splits (Seitzer et al., 2022). We also considered multiple experiments per dataset to minimize the impact of randomization. The dataset size and input/output dimensions of each problem are listed in Table 3 and Table 1 under each column within parentheses. We report the metrics at the original scale and averaged over all outputs.

- **Architectures:** We use a one-layer MLP with 50 neurons followed by *ReLU* activation function for all ME and MVE methods. For the proposed cooperative learning strategy, we employ the same architecture for mean net and BNN, and we additionally use an MLP with 5 neurons followed by *ReLU* activation function as the variance network.

---

[5] We also try the same architecture with other approaches; however, BBB has difficulty with such a large MLP architecture

- **Optimizer/Inference:** The methods, including ME, MVE, BBB, as well as the warm-up of the proposed cooperative learning strategy, are trained via *Adam* for 20000 epochs with a learning rate of 0.001 (0.01 for training ELBO), in which the MC-Dropout has a Dropout rate of 0.1 for each layer. As for the pSGLD, we set the burn-in epoch to be 5000 and collected 150 posterior samples every 100 epochs (In total 20000 epochs, which is the same as that of *Adam*). For the additional variance network of the proposed cooperative learning strategy, we use *Adam* to train for 10000 epochs with an early stopping of 100 epochs. The batch size is set to be 256 for all approaches.

- **Hyperparameter selection:** We split each dataset into train-test by 80%-20% randomly 20 times. In addition, the train set is further divided into 80%-20% for training and validation. We search for the best learning rate within $\{10^{-4}, 3 \cdot 10^{-4}, 10^{-3}, 3 \cdot 10^{-3}, 7 \cdot 10^{-4}\}$ and as well as record the best epoch utilizing the validation NLL loss. After finding the best learning rate and epoch, the final model is trained using all training data points, and metrics are calculated for the test set. It is noted that the presented results of MVE ($\beta$-NLL) in Table 3 and Table 1 are the overall best results among $\{\beta = 0.0, \beta = 0.25, \beta = 0.5, \beta = 0.75, \beta = 1.0\}$. We also conducted a grid search for BNN-Homo methods for suitable constant noise, where the space is set between zero and one, with 10 different values given the consideration of similar computational resources. For the proposed cooperative learning strategy, we feed all the data to Algorithm 1 and set the iteration $K = 2$.

### E.3 IMAGE REGRESSION DATASETS

We take the image regression dataset introduced in (Gustafsson et al., 2023), which consists of relatively large-scale problems, with each containing between 6,592 and 20,614 training images depending on the specific task. Each image has a resolution of $64 \times 64$ and the target is a one-dimensional output $y$. Full details can be found in (Gustafsson et al., 2023); a summary of the relevant information is provided below:

- **Cells** The dataset contains 10000, 2000, and 10000 images for training, validation, and testing, respectively. The labels have a range of $[0, 200]$, and there is no distribution shift among all the datasets.

- **Cells-Tail** The training and validation datasets contain images with labels in the range of $[50, 150]$; the test dataset has labels with a range of $[0, 200]$

- **Cells-Gap** The training and validation datasets contain images with labels in the range of $[0, 50] \cup [150, 200]$; the test dataset has labels with a range of $[0, 200]$

- **ChairAngle** The dataset 17640, 4410, and 11225 images for training, validation, and testing, respectively. The labels have a range of $[0.1°, 89.9°]$, there is no distribution shift among all the datasets.

- **ChairAngle-Tail** The training/validation datasets contain images whose labels have a range of $[15°, 75°]$; and the test labels have a range of $[0.1°, 89.9°]$.

- **ChairAngle-Gap** The labels have a range of $[0.1°, 30°] \cup [60°, 89.9°]$, and the test labels are in $[0.1°, 89.9°]$.

- **SkinLesion** The dataset contains 6592, 1164, and 2259 images for training, validation, and testing, respectively. The dataset contains four different sub-datasets, in which the first three are split into train/val with $85\%/15\%$; and the fourth sub-dataset is used as the test dataset.

- **AreaBuilding** The dataset contains 180 large aerial images with corresponding building segmentation masks. Specifically, the train/val is obtained from two densely populated American cities cities while the test dataset is from rural European cities. Overall, it contains 11184, 2797, and 3890 images for training, validation, and testing, respectively.

We leverage the hyperparameters in (Gustafsson et al., 2023) and define ours as follows:

- **Architectures:** ResNet34 backbone (He et al., 2016) is employed for this problem. We use a two-layer MLP to decode the prediction into a Gaussian distribution for MVE ($\beta$-NLL),

MVE (Ensembles), and MVE (MC-Droout). It is noted that a dropout layer with a dropout rate of 0.1 is followed for each MLP layer. As for the variance net and deep evidential regression, the decoding layer is set to be the corresponding outputs after the ResNet34 backbone.

- **Optimizer/Inference** We use *Adam* with a learning rate of 0.001 for MVE, as well as the warm-up step of the proposed cooperative learning strategy, and employ *Adam* to run for 75 epochs for the above methods witha batch size of 32. As for the pSGLD, we set the burn-in epochs to be 20 and we sample 10 posterior samples every 2 epochs (in total 40 epochs). As for the additional variance network, which is trained with *Adam* for 20 epochs.

### E.4 PLASTICITY LAW DATASETS

**Hyperparameter setting for Section 4** In the literature of data-driven constitutive laws, several studies address similar problems without considering noise (Anonymous, 2025). We leverage their setups and define the hyperparameter setting as follows:

- **Architectures:** For the mean network and BNN, we adopt a two-layer GRU architecture with 128 hidden neurons. It is noted that we only apply the Dropout operation to the hidden-to-decoding layer with a Dropout rate of 0.02. The reason is that this inference method does not show compatible performance when making all weights and biases the Dropout layer. For the proposed cooperative learning strategy, a smaller GRU network with two layers and 8 hidden neurons is employed for the variance network.

- **Optimizer/Inference** We use *Adam* with a learning rate of 0.001 for ME, MVE, as well as the warm-up step of the proposed cooperative learning strategy and employ *Adam* to run for 2000 epochs for the above methods. As for the pSGLD, we set the burn-in epoch to be 500 and we sample 150 posterior samples every 10 epochs (in total 2000 epochs). The additional variance network is trained with *Adam* for 4000 epochs with an early stopping patience of 50 epochs.

- **Hyperparameter selection:** For every experiment of different training points we reserve 100 validation data points from the training dataset to determine the best epoch for ME and MVE. Subsequently, we combine the validation points with the training set and retrain the final model using the best epoch configuration. For the same reason, the results depicted of MVE ($\beta$-NLL) is the overall best among $\{\beta = 0.0, \beta = 0.25, \beta = 0.5, \beta = 0.75, \beta = 1.0\}$. As for the BNN-Homo, we follow the same strategy to execute a grid search for the best homoscedastic noise, where the noise variance is set to be $\{0.04, 0.06, 0.08, 0.10, 0.12\}$ empirically. For the proposed cooperative learning strategy, we set the iteration $K = 2$

**Hyperparameter setting for Appendix D.6** To investigate the influence of the variance network architecture, we consider eight configurations where the number of layers varies from 1 to 2, and the number of hidden neurons is set to $\{8, 16, 64, 128\}$. The largest configuration, $\{128, 2\}$, is identical to the mean network. All other hyperparameters are consistent with those used in the previous section.

### E.5 COMPUTATIONAL RESOURCES

Each experiment, except the image regression datasets, is conducted on a node of an HPC cluster platform with an Intel® Xeon(R) E5-2643v3 CPU with 6 cores of 3.40GHz and 128 GB of RAM. Regarding the image regression datasets, the experiment is conducted based on a platform with an H100 NVL GPU.

# F    DESCRIPTION OF PLASTICITY LAW DATASET

The fundamental mechanical law of materials is called a constitutive law. It relates average material deformations to average material stresses at any point in a structure. Constitutive laws can model different Physics behaviors, such as elasticity, hyperelasticity, plasticity, and damage. In this paper, we focus on generating datasets for plastically deforming composite materials, coming from prior work (Anonymous, 2025). Without loss of generality, the constitutive law of such path-dependent materials can be formulated as follows:

$$\mathbf{y} = f(\mathbf{x}, \dot{\mathbf{x}}, \tau, \dot{\tau}, \boldsymbol{h}) \tag{54}$$

where $\mathbf{y}$, $\mathbf{x}$, $\tau$ are stress, strain, and temperature respectively, $\boldsymbol{h}$ is a set of internal variables. The constitutive law can be predicted by micro-scale simulations of material domains that are called stochastic volume elements (SVEs) – see Figure 23. These SVEs are simulated by rigorous Physics simulators based on the Finite Element Method (FEM). Each material SVE is utilized as the basic simulation unit. Many factors bring data uncertainty into the data generation process; we focus on data uncertainties from two aspects: (1) SVE size; and (2) particle distribution. As we randomize particle distribution, the stress obtained for an input deformation exhibits stochasticity (aleatoric uncertainty). Therefore, two datasets are generated from simulations according to Table 10.

Table 10: Parameter configuration material plasticity law simulations (Units: $\mathrm{SI}(mm)$ ).

| Name | Microstructure Parameters | | | Hardening Law | $E_{\mathrm{fiber}}$ | Size | $E_{\mathrm{matrix}}$ | $\nu_{\mathrm{matrix}}$ | $\nu_{\mathrm{fiber}}$ |
| --- | --- | --- | --- | --- | --- | --- | --- | --- | --- |
| | $v_f$ | $r$ | $r_{\mathrm{std}}$ | | | | | | |
| Material 1 | 0.30 | 0.003 | 0.0 | $\sigma_y = 0.5 + 0.5(\bar{\epsilon})^{0.4}$ | 1 | 0.048 | 100 | 0.30 | 0.19 |
| Material 2 | 0.30 | 0.003 | 0.0 | $\sigma_y = 0.5 + 0.5(\bar{\epsilon})^{0.4}$ | 1 | 0.030 | 100 | 0.30 | 0.19 |

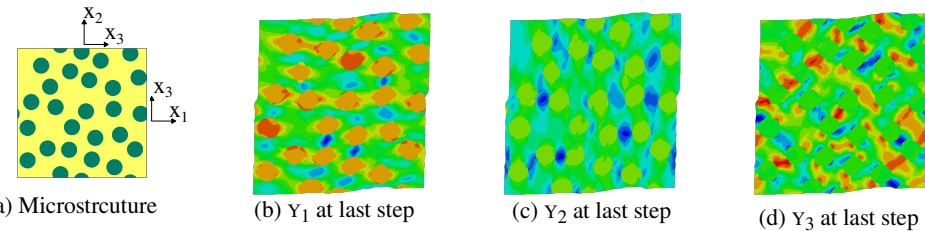

(a) Microstrcuture     (b) $Y_1$ at last step     (c) $Y_2$ at last step     (d) $Y_3$ at last step

Figure 23: **Material plasticity law simulation illustration.** Figure. (a) shows an arbitrary realization of material microstructure, and the following figures show the contour plot of this material simulation at the final step.

According to Table 10, we have two materials that have different SVE sizes that control the noise sources of the data. Materials with a smaller SVE size have larger data noise. Figure 23 and Figure 24 illustrate details of the simulations and how the uncertainty in the data originates. Specifically, according to the microstructure configuration in Table 10, we generate SVEs using the Monte Carlo Sampling strategy (Melro et al., 2008) and simulate the stress responses through the commercial software ABAQUS (Dassault Systèmes, 2024) with the input of the strain sequence shown in the first row of Figure 24. After simulation, we get a series of contours of stress components in Figure 23 and average the field (color contours) for each input sequence point, leading to the output sequence. An example of 3 realizations of SVEs and corresponding 3 output response sequences, shown as a dashed line in the second row of Figure 24. Each realization corresponds to a randomization of the microstructure of the material (particle distribution). By running multiple realizations, we can obtain the statistics of those strain inputs. By definition, the variation in the stress output is the uncertainty of the data.

Each input deformation sequence and output stress sequence used in training contains 100 points. In total, we use 50 different SVEs to ensure that we have enough realizations to calculate the ground truth aleatoric uncertainty. Overall, we simulate 1000 sequences for training and 100 sequences for testing, respectively, for each problem shown in Table 10.

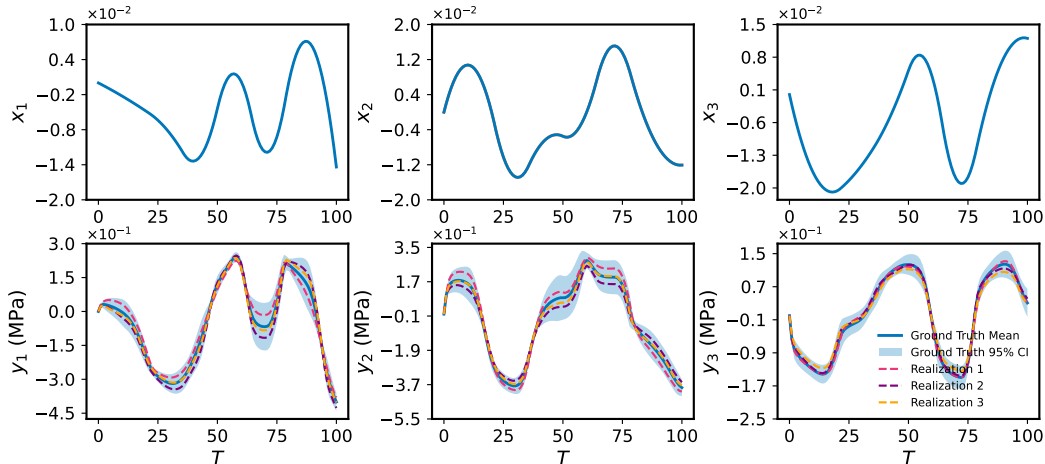

Figure 24: **Plasticity law data illustration.** The first column is the strain inputs for the material law simulation, and the second row is the stress outputs. Each dashed line represents one particle material microstructure realization in Figure 23; the mean and the confidence interval are obtained via multiple realizations.

The new dataset is made available as open-source in the hope of creating a more interesting problem for assessing future methods because we had difficulties in finding more challenging heteroscedastic problems with ground truth aleatoric uncertainty to assess our method. This dataset is three-dimensional and history-dependent, i.e., $\mathcal{D} = \{\mathbf{x}_{n,t}, \mathbf{y}_{n,t}\}$ with features $\mathbf{x}_{n,t} \in \mathbb{R}^3$ and targets $\mathbf{y}_{n,t} \in \mathbb{R}^3$, where $n = 1, ..., N$ are the training sequences (deformation paths) and $t = 1, ..., T$ are the points in each sequence. We highlight two aspects about this dataset. First, the targets $\mathbf{y}$ are history-dependent, so estimating a new state $\mathbf{y}'$ requires to know the sequence of states needed to reach that state, i.e., regression requires recurrent neural network architectures (Anonymous, 2025), specifically adopting a Gated Recurrent Unit (GRU) architecture (Cho, 2014; Gan et al., 2017) in this work. Furthermore, the dataset was created synthetically by physically-accurate computer simulations of materials under mechanical deformation, so it was possible to generate enough data to determine the ground-truth aleatoric uncertainty (arising from variations within the material). In other words, we have a good estimate of the heteroscedastic noise in the data.

