# OpenReview forum: "Cooperative variance estimation and Bayesian neural networks disentangle aleatoric and epistemic uncertainties"
_ICLR.cc/2026/Conference — Submitted to ICLR 2026_

### Official Review · Reviewer_gD7g · 2025-10-16

**Soundness:** 2
**Presentation:** 2
**Contribution:** 2
**Rating:** 2
**Confidence:** 4

**Summary:**

The paper focuses on Bayesian heteroscedastic regression with the goal to disentangle the estimate for the variance (aleatoric) and the model uncertainty. The proposed method consists of three steps in which first the mean is estimated and subsequently the variance and the BNN are updated iteratively for K steps. The marginal likelihood is approximated with the MAP estimate of a Gaussian likelihood and a Gaussian prior. The main novelty of the method lies in the loss function for the variance estimation that is based on the squared residuals, which the authors show are distributed according to a Gamma distribution.

**Strengths:**

- The proposed approach is evaluated on a large range of tabular and image regression datasets.
- An interesting synthetic dataset based on plasticity laws is introduced, however, it is unclear if this is a contribution of the submitted paper, or of Anonymous (2025).

**Weaknesses:**

- The authors write that “no current method achieves reliable uncertainty disentanglement”, referring to Amini et al. (2020) and Immer et al. (2023) by citing Mucsányi et al. (2024). After briefly checking the latter paper, it appears, however, that it does not cite Amini and Immer. Thus, this statement is incorrect in this context. The same argument applies for the statement made at the end of Sec. 3.2, which is confusing. Immer et al. (2023) estimate both the aleatoric and epistemic uncertainty without introducing a new hyperparameter. This approach is also omitted as a baseline although it achieves stronger results on the UCI regression tasks, e.g., on concrete, energy, boston, wine, and yacht.
- The formal justification of the method is weak (see questions).
- Seitzer et al should be cited after Eq. (3).
- In Sec 3.3.1 the authors should state a formal proposition or lemma that they proof.

**Questions:**

What theoretical advantage is there to train three instead of a single network? Since the authors advocate to disentangle the networks for the two parameters (mean and variance), how does this justify having to learn two output parameters for the variance network alone? Does the corresponding loss have desirable properties?

---

> ### Author Response · Authors · 2025-11-28
> **Responses to the reviewer**
>
> **[W1]: Incorrect statement, and missing baseline of Immer et al. (2023).**
>
> Thank you for pointing this out. We note that the results reported in Immer et al. (2023) are not directly comparable to ours because their method relies on per-layer hyperparameter tuning of the neural weight priors (via empirical Bayes or grid search), whereas our setup adopts a simple Gaussian prior that is shared across all layers for all baselines. As a consequence, the numbers published in their paper cannot be used as a plug-in baseline under our evaluation protocol.
>
> To address the reviewer’s concern fairly and transparently, we have re-ran the publicly available implementation of Immer et al. (2023) under our unified evaluation pipeline, in which:
> - All methods, including Immer et al., are trained on the same training set;
> - Predictive uncertainties are calibrated using the same validation procedure (following `Reviewer aT67`’s suggestion);
> - We report RMSE, TLL, test coverage, and interval length on the test set.
>
> The results are given as follows:
>
> | Metric | Method | carbon | concrete | energy | boston | power | superconduct | wine-red | yacht |
> |--------|--------|--------|----------|--------|--------|--------|---------------|-----------|--------|
> | **RMSE ($\downarrow$)** | Immer et al. (2023) | 0.0072 (0.0026) | 5.24 (0.68) | 1.06 (0.32) | 3.85 (1.48) | 3.84 (0.15) | 13.94 (5.41) | 0.64 (0.05) | 1.39 (1.58) |
> | | **VeBNN (pSGLD)** | **0.0064 (0.0030)** | **4.70 (0.77)** | **0.77 (0.12)** | **3.64 (0.82)** | **3.79 (0.14)** | **12.23 (0.45)** | **0.63 (0.05)** | **0.87 (0.35)** |
> | **TLL ($\uparrow$)** | Immer et al. (2023) | 3.88 (0.24) | -3.07 (0.16) | -1.12 (0.15) | -2.74 (0.37) | -2.76 (0.04) | -3.47 (0.12) | -0.96 (0.07) | -1.69 (1.84) |
> | | **VeBNN (pSGLD)** | **4.04 (0.78)** | **-2.87 (0.18)** | **-0.85 (0.10)** | **-2.58 (0.21)** | **-2.74 (0.04)** | **-3.41 (0.09)** | **-0.95 (0.08)** | **-1.27 (0.57)** |
> | **Test Coverage ($\rightarrow 0.95$)** | Immer et al. (2023) | 0.9497 (0.0063) | 0.9379 (0.0411) | 0.9396 (0.0262) | 0.9324 (0.0552) | 0.9463 (0.0079) | 0.9506 (0.0076) | 0.9459 (0.0312) | 0.9306 (0.0680) |
> | | **VeBNN (pSGLD)** | 0.9473 (0.0064) | **0.9398 (0.0388)** | **0.9532 (0.0184)** | **0.9382 (0.0552)** | **0.9492 (0.0078)** | **0.9511 (0.0068)** | **0.9528 (0.0284)** | **0.9323 (0.0644)** |
> | **Test Interval Length($\downarrow$)** | Immer et al. (2023) | 0.0213 (0.0016) | 20.31 (3.24) | 3.13 (0.68) | 13.07 (5.27) | 14.59 (0.54) | 39.27 (1.92) | 2.51 (0.24) | 2.43 (1.16) |
> | | **VeBNN (pSGLD)** | **0.0141 (0.0011)** | **17.71 (2.93)** | **2.45 (0.27)** | **12.60 (3.44)** | **14.66 (0.53)** | **39.94 (2.00)** | **2.66 (0.26)** | **2.71 (0.68)** |
>
> Overall, VeBNN (pSGLD) achieves consistently better RMSE and TLL across all datasets. For several datasets (carbon, concrete, energy, boston), VeBNN attains similar test coverage while yielding sharper (shorter) predictive intervals; for the remaining datasets (power, superconduct, wine-red, yacht), VeBNN has comparable coverage and interval length but improved mean predictions, resulting in better TLL.
>
> **[W2, Q1]: Justification of  using two networks for uncertainty disentanglement  comparing with using one network with two parameters.**
>
> Motivated by this comment, we worked very hard to create theoretical arguments that would prove the clear advantage of our method, beyond the extensive empirical evidence provided with the 18 datasets we considered. We added Appendix A.2 and A.3 to demonstrate the theoretical advantage of the cooperative strategy. Please, also see the answer to Reviewer J3sN concerning this point. In summary, the loss of the jointly trained MVE network has saddle points, therefore it is possible to have zero gradients with respect to the mean and the aleatoric uncertainty but not being at an optimum. This explains the characteristic trade-off that we see across the 18 datasets when using joint training, and that is illustrated in Fig. 1. Furthermore, note that recent literature also empirically reported similar observations. Fortunately, now we also provide theoretical arguments that justify them and, in our opinion, the paper is significantly strengthened.
>
>
> **[W3] Seitzer et al should be cited after Eq. (3).**
>
> Thanks for the suggestion. We cited Seitzer et al after Eq. (3).
>
> **[W4] In Sec 3.3.1 the authors should state a formal proposition or lemma that they proof.**
>
> Thanks for the suggestion, we added “Lemma 3.1. We aim to demonstrate that from Assumption 3.1 and for y following a Gaussian distribution, the squared residual r = (μ(x; θ) − y)² follows a Gamma distribution.”

---

### Official Review · Reviewer_Zmnw · 2025-10-21

**Soundness:** 2
**Presentation:** 1
**Contribution:** 2
**Rating:** 4
**Confidence:** 4

**Summary:**

The paper introduces a novel approach to separate aleatoric and epistemic uncertainty in MVE networks. The method proceeds in three stages: first train the network to estimate the mean; then train a separate variance network to model the aleatoric (data) uncertainty; and finally train a Bayesian neural network to estimate epistemic (model) uncertainty. The authors demonstrate the approach on a toy heteroscedastic regression problem, several UCI benchmark datasets, and large-scale image regression tasks.

**Strengths:**

1. The method is straightforward to implement and scalable to different architectures.

2. The paper is well-written and easy to follow.

**Weaknesses:**

1. The paper introduces no novel theoretical contribution. The proof of assumption 3.1 is a known result.

2. The experimental results are confusingly presented. It is unclear whether the values in parentheses in Table 1 represent standard deviations. If so, the reported overlaps between methods suggest that performance differences are not statistically significant, calling into question the claimed improvements. Additionally, some acronyms in the results are undefined such as TLL.

3. The model architectures (two hidden layers with 256 neurons each) seem prone to overfitting, particularly on small or low-complexity datasets such as the toy sinusoidal example. This raises doubts about the robustness and generalizability of the proposed method.

4. The paper only addresses Gaussian (homoscedastic) aleatoric noise and evaluates a single type of neural architecture. This narrow focus limits the applicability and generality of the proposed uncertainty separation compared to related works [1, 2].

5. Several plots in Figure 2 and 5 are missing lines from the legend. These presentation issues make it difficult to interpret the qualitative results and reproduce the experiments.

6. The paper reports RMSE and TLL but omits standard uncertainty quality measures such as AUROC for out-of-distribution detection, limiting interpretability of the claimed uncertainty improvements.


[1] Berry, Lucas, and David Meger. "Normalizing flow ensembles for rich aleatoric and epistemic uncertainty modeling." Proceedings of the AAAI Conference on Artificial Intelligence. Vol. 37. No. 6. 2023.

[2] Berry, Lucas, and David Meger. "Efficient epistemic uncertainty estimation in regression ensemble models using pairwise-distance estimators." arXiv preprint arXiv:2308.13498 (2023).

**Questions:**

1. Why was method not applied to Ensembles (MVE)? I imagine this would work as it did for MC Dropout.

2. Do you have the RMSE as a value instead of percentage? It would be nice to see this in the paper, maybe in the appendix.

---

> ### Author Response · Authors · 2025-11-28
> **Responses to weaknesses**
>
> **[W1]:  Proof of assumption 3.1 is a known result.**
>
> Indeed, we use this known proof to justify that the Gamma log likelihood loss can be derived for the squared residual. This allows to use this loss to find the aleatoric uncertainty, improving the estimate when compared to the Gaussian NLL loss  in Eq. (2). The novelty of VeBNN is the new training strategy, which leads to stable optimization by executing step 1 and 2 separately. Then, we can carry out inference on a fixed aleatoric model that further improves the performance while giving epistemic uncertainty. We will clarify this positioning in the revision.
>
> **[W2]: Experimental results are confusingly presented in Table 1.**
>
> We apologize for the missing explanations. The numbers in parentheses denote the standard deviation over 20 runs, and “TLL” refers to test log-likelihood, where higher is better.
>
> The UCI regression datasets provide only a single realization per test point and no ground-truth aleatoric uncertainty. Therefore, we can only report TLL using mean and total predictive variance, which inherently underestimates our model’s ability, i.e., the uncertainty disentanglement. Following the suggestion from `Reviewer aT67`, we include two extra metrics: test coverage and test interval length, as well as calibrate the total predictive variance for all methods in order to have a fair comparison. After that, we observe that the VeBNN leads to even better improvements in terms of TLL, and  produces substantially sharper and better-calibrated predictive intervals compared to the baselines.
>
> **[W3]: Architecture choice of the illustrative example.**
>
> Thank you for the comment regarding the neural architecture (two hidden layers with 256 neurons each). We would like to clarify that this architecture is used only in the one-dimensional illustrative example, and although it is relatively large, Appendix D.1 presents extensive experiments across different training sizes. As shown in Figures 11 and 13, our method does not exhibit overfitting.
>
> In addition, the mean-network architectures are chosen according to standard practice in each domain: one-layer feedforward neural network with 50 neurons for the UCI regression datasets,  a convolutional neural network (ResNet34) for the image regression datasets, and a recurrent neural network with two-layer GRU architecture for the plasticity law discovery datasets. All architectures are taken from corresponding literature. Moreover, we also investigate the influence of the variance estimation network in Sec. 5, and results in Figure 19 shows that the proposed VeBNN is also robust to the size of the variance estimation network.
>
> **[W4]: Only consider homoscedastic aleatoric noise and only a single neural architecture, with limited applicability and generality comparing with [1, 2].**
>
> We would like to clarify that there is a misconception about this point. In fact, we considered three types of architectures (FNN, CNN and RNN), and we considered heteroscedastic aleatoric noise in all main experiments (UCI regression, image regression, and plasticity law discovery). Modeling unknown heteroscedastic aleatoric uncertainty is the core motivation of our proposed method, and why we think that the scope is quite broad. We also made the effort to consider examples with 3 different architecture types to prove this point.
>
> Thank you for the references. They are interesting and, despite having different scope, they are complementary to what we are proposing. They consider input aleatoric uncertainty and investigate uncertainty propagation through a nonlinear functional transformation. Our interests are in estimating unknown heterosceadtic aleatoric uncertainty present on the outputs, and to disentangle it from the epistemic uncertainty.
>
> **[W5]: Missing lines from the legend in Figures 2 and 5.**
>
> Thank you for examining Figures 2 and 5 in detail. Here’s the clarification:
>
> *Figure 2:*
> - The left subfigure (RMSE) evaluates mean prediction, so all methods are included and all eight lines appear.
> - The middle subfigure (Epistemic TLL) evaluates epistemic uncertainty; therefore, methods without epistemic modeling capability, ME (MSE) and MVE (β-NLL=0.5), do not appear.
> - The right subfigure (WA) evaluates aleatoric uncertainty prediction; ME(MSE) is absent for the same reason. The line for Evidential is partially cut off due to its scale, and MVE (MC-Dropout) does not appear because its predicted aleatoric variance is extremely large and lies outside the plot range.
>
> *Figure 5:*
> - The third subfigure misses the line of using MSE loss for End-to-End training of BNNs, since it lacks the ability of predicting aleatoric uncertainty. We clarify this in the revised version.

---

> ### Author Response · Authors · 2025-11-28
> **Responses to questions**
>
> **[W6]: Omit AUROC for out-of-distribution detection.**
>
> Since our focus is regression rather than classification-based OOD detection, AUROC is not directly relevant. Instead, we evaluate test coverage and interval length under distribution shift. Specifically, for regression-type distribution shift, we have the image regression datasets: we train the model under one distribution and evaluate its behavior under a shifted distribution using test coverage and the corresponding predictive interval length.
>
> **[Q1]: Why was method not applied to Ensembles (MVE)?**
>
> Thank you for this point. We have now included ensembling in our method. This also helps strengthen the point that the cooperative strategy improves performance compared to joint training.
>
>
> **[Q2]: Do you have the RMSE as a value instead of percentage?**
>
> Thank you for the question. We report RMSE as a percentage only for the plasticity law discovery datasets. Each data point contains three sequences of 100 steps (Figure 21), and their magnitudes differ. Using absolute RMSE would therefore overweight the larger channels and undervalue errors on others. For this reason, we use the normalized RMSE defined in Eq. (28) to provide a fair comparison across all three outputs.

---

### Official Review · Reviewer_aT67 · 2025-11-01

**Soundness:** 2
**Presentation:** 2
**Contribution:** 3
**Rating:** 4
**Confidence:** 4

**Summary:**

The submission suggests novel procedural improvements on how to estimate aleatoric and epistemic uncertainty for regression by modifying training algorithms of Bayesian Neural Networks (BNN). Extensive empirical experiments are presented. These procedural improvements sometimes lead to significantly better performance than plain BNNs.

These procedural improvements are:

* A: Step 1: a MAP as a mean predictor, then Step 2: fit the aleatoric uncertainty to the residuals (keeping the mean from Step 1 fixed for computing these residuals). Then Step 3: train a BNN with fixed aleatoric uncertainty from step 2.
* B: They modify Step 2 of A by using a Gamma distribution
* C: They modify Step 2 of A by using a more restricted (i.e., smaller) architecture for Step 2 than for the other steps.
* D: In step 3, they warm-start the BNN with the MAP from step A (not novel, but a reasonable thing to do)

**Strengths:**

Evaluating the methods on 18 datasets (including multiple real-world datasets) and adding some ablation studies is very valuable. Not only is it interesting to see the performance of your procedural improvements, but for me, it was also interesting to see the performance comparison between multiple other methods.

The improvement achieved with your procedural improvements is very impressive for some of the datasets.

Uncertainty quantification is an important problem for many applications. In particular, epistemic and aleatoric uncertainty.

Training BNNs can be very challenging, so improving their training with some procedural improvements is very valuable.

The material plasticity datasets with known aleatoric uncertainty could be very valuable, also for benchmarking other methods that disentangle aleatoric and epistemic uncertainty for regression. Usually, real-world datasets are more important than synthetic ones, but this one seems like a realistic (practical?) simulated dataset. I have no expertise in material science, but at first sight it seems legit. Do any of the other reviewers better understand the relevance of this dataset?

I like Figure 2. In particular, the Wasserstein distance between the true aleatoric and the estimated aleatoric. This is something that wouldn’t be possible for most real-world datasets.

**Weaknesses:**

1. The performance of the results varies quite a lot. While for some combinations of datasets and models, your procedural improvements improve the performance, for others, your procedural improvements worsen the performance. Overall, I think your procedural improvements have, on average, a rather positive impact, and therefore I think it is a very valuable contribution, but I am not sure if it meets the bar of ICLR. My impression is that for the materials, your procedural improvements are very consistently helpful; for the UCI datasets, the procedural improvements are, on average, more helpful than harmful, but for the deep learning image datasets, the results are much harder to interpret. For the image regression datasets, the classical ensembles are also very strong, and often better than your BNNs.

2. Metrics: Why do you show different metrics for different datasets? You should also show TLL (and VLL) for the deep learning datasets (in the appendix). You should also show interval length and coverage for the UCI datasets (in the appendix). Pinball loss would also be an interesting addition of another suitable metric, but your choice of metrics is already OK if you apply them across all datasets. Maybe you should also show the test interval length? (The validation coverage is equal for all methods due to calibration on the validation dataset?) I think the test interval lengths when calibration on the validation dataset would be interesting, as this is what you would obtain in practice, but also the test interval length when calibrated on the test dataset would be interesting, to make the numbers comparable. Otherwise, if one method has worse coverage but better length, they are hard to compare. But I think I see your point that the test interval lengths might be hard to compare, as the test coverages of different methods vary a lot.

3. Calibration: In practice, uncertainty should (almost) always be calibrated on the validation dataset or some separate calibration dataset. The current version of the submission is quite unclear about when you are calibrating your uncertainties. Please be very precise about this in the revision. In the main paper, you never mention any form of calibration. In Eq.~(33) on page 17 in the appendix, you mention that you are using a constant additive calibration constant that you calibrate on the validation dataset. However, it seems as if you only calibrate your uncertainty for the coverage/length metrics? Are you using the uncalibrated uncertainties for the TLL experiments? How do you choose κ? κ strongly affects the scale of the epistemic uncertainty. Especially if you don’t do any hyperparameter optimization (HPO), you really should use some calibration. E.g., you can do classical calibration (https://arxiv.org/abs/1706.04599, https://arxiv.org/abs/1807.00263, https://www.mdpi.com/1424-8220/22/15/5540) by multiplying your total predictive variance from Eq. (11) by a constant $c$, either such that it maximizes the VLL (Validation Log Likelihood) or such that the 95% predictive interval covers 95% of the validation data. (Alternatively, you could try something more recent like https://arxiv.org/abs/2507.08150 where you use  2 calibration constants: multiply $c_1$ on aleatoric variance and $c_2$ on the epistemic variance, and optimize these two constants by maximizing the VLL (constrained on your predictive interval covering $1-\alpha$ of the validation data).) Maybe better calibration (e.g., multiplicative instead of additive, or 2 constants instead of 1) would also improve your test coverage? Multiplicative calibration has the advantage that it is compatible with metrics such as TLL (scaling up the standard deviation of a Gaussian by a constant $c$ still results in a Gaussian). I think there is quite a high chance that the TLL can be improved by calibrating via the VLL. Maybe this would mitigate the seemingly random differences in performance rankings of the methods across different datasets.

4. Math: When I try to derive a formula for the NLL of a Gamma Distribution, I obtain a different result. Are you also using this density for the Gamma distribution: $f(x)={\frac {\lambda ^{\alpha }}{\Gamma (\alpha )}}x^{\alpha -1}e^{-\lambda x}$? When I compute the logarithm of this I obtain  α Log[λ] -Log[Gamma[α])] + (α - 1)Log[x] - λx, which I have checked with WolframAlpha: https://www.wolframalpha.com/input?i=Log%5B%28%28%CE%BB%5E%CE%B1%2FGamma%5B%CE%B1%5D%29+x%5E%28%CE%B1+-+1%29%29%2FE%5E%28%CE%BB+x%29%5D+-+%28++%CE%B1+Log%5B%CE%BB%5D+-Log%5BGamma%5B%CE%B1%5D%29%5D+%2B+%28%CE%B1+-+1%29Log%5Bx%5D+-+%CE%BBx++%29
I don’t understand, for example, why you divide by $r_n$ in the last term instead of multiplying by $r_n$? The first term also seems incorrect to me. Can you please clarify?

5. Why don’t you apply your procedural improvements to deep ensembles as well? They can also be seen as a simple approximation of BNNs.

6. There are also some issues with the presentation, which is sometimes not clear (see Questions for detailed feedback).

**Questions:**

Q1: Lines 36-37: The sentence “In such cases, an outcome with good mean performance but large aleatoric uncertainty may be unacceptable.” feels confusing. Maybe write something weaker, like “undesirable” instead of "unacceptable"? And/or maybe formulate it the other way around: “In such cases, an outcome with bad mean performance but low aleatoric and low epistemic uncertainty might be unacceptable.”? What exactly do you mean by “mean performance”? Or maybe skip the sentence?

Q2: Lines 37-38: The next sentence, “Therefore, the principle of reducing epistemic uncertainty behind active learning or decision-making needs to be balanced by the respective prediction of aleatoric uncertainty.” feels even more confusing. How does it need to be balanced? What does this mean? Or maybe skip the sentence?

Q3: Lines 37-38: Why not cite (Lakshminarayanan et al., 2017) for deep ensembling?

Q4: Lines 128-129: Maybe the sentence: “Still, a recent investigation (Mucsányi et al., 2024) has shown that no current method achieves reliable uncertainty disentanglement.” should be weakened? Maybe “suggests” instead of “has shown”. Is it a theoretical result or an empirical one? “No current method” sounds very hard to prove.

Q5: Section 3.1: I think the more typical notation would switch the variable names $s$ and $\varepsilon$.

Q6: Line 162: “with of” is a typo? Should be “of”?

Q7: Line 186: Important question: In Step 2, when you are using the fixed mean from Step 3, are you using the MAP or the mean posterior from Step 3?.

Q8: Line 191: Instead of $\theta^*$, the estimated posterior distribution over $\theta$ is meant? Or actually only the MAP?

Q9: Line 194: Concretely, you are setting constant=1 for Step 1?

Q10: Line 200: Instead of writing: “There is, however, an important detail” write "There are, however, multiple important details"

Q11: Line 203: Instead of writing “the variance network outputs the residual” I would write something like, “the variance network estimates the distribution of the squared residuals” or “the variance network gets the squared residuals […] as training-labels”.

Q12: Lines 203-204:The Gamma distribution falls a bit from the sky. I would add “, as explained in Lemma 3.1.” at the end of the sentence.

Q13: Lines 208-210: The structure is very confusing. A proof should always follow a lemma/theorem/proposition/corollary. Therefore, I would write “Lemma 3.1. We aim to demonstrate that from Assumption 3.1 and for y following a Gaussian distribution, the squared residual r = (μ(x; θ) − y)² follows a Gamma distribution.
Proof. Since [...]”

Q14: Eq.~(6-7) seems mathematically wrong to me. Please clarify.

Q15: Lines 256-257: For “From experience, preconditioned Stochastic Gradient Langevin Dynamics (pSGLD) (Li et al., 2016) is expected to be a good choice for this BNN because the aleatoric uncertainty is fixed, and the likelihood and prior are both Gaussian, leading to a Gaussian posterior.” a short explanation of “preconditioned” might be nice if you have space left? More importantly, what do you mean by Gaussian posterior? Gaussian predictive distribution (or the posterior of µ is Gaussian)? Or the posterior on the parameters is Gaussian? Why is this Gaussian? Do you mean the true posterior of the theoretical BNN prior or the posterior approximated by pSGLD?

Q16: Line 301: The abbreviation "TLL" is only introduced in the appendix.

Q17: Line 318: In the last column, “-0.64 (0.28)” should be bold.

Q18: Line 323: “The conclusions are similar to those obtained from the UCI regression datasets.” is too simplified, I think. The results are less clear there, I would say.

Q19: Line 332: I don’t agree with the end of the sentence: “This is very encouraging because training is faster for  VeBNN (pSGLD) when compared to MVE (Ensembles), as the former collects samples in the same training procedure, while ensembling requires collecting one sample per training of a single MVE (i.e., training restarts many times).”. Deep Ensembles can be parallelized much more easily, as each ensemble member can be trained in parallel. For Deep Ensembles, usually fewer epochs are needed compared to pSGLD. Therefore, I definitely would not say that pSGLD is faster. I think Deep Ensembles is faster if you parallelize. Maybe you can say that pSGLD has lower computational costs, but even there, I am not sure. VeBNN also adds sequential computational costs, which cannot be parallelized. This also makes it slower, especially when K increases.

Q20: Line 368-369: In “(upper column to bottom column)”, it should be “row” instead of “column”.

Q21: Lines 403-405: I have no idea what “for 800” refers to in “To further explore this, we show the trajectory of all essential components of VeBNN with respect to the iteration K from 1 to 5 for 800 in Figure 18.”. Can you clarify?

Q22: Lines 483-484: Important question: What do you mean by “As a  Bayesian method, it does not require validation data.”? Aren’t you doing HPO on the validation dataset? Aren’t you calibrating your uncertainty on the calibration dataset? I personally think that for basically any ML task, a validation dataset is strongly recommended. Your method has many hyperparameters, so I really think you should use a validation dataset, and I think, in fact, you are using one?

Q23: Line 915: “Should it be “predictive interval” instead of “confidence interval'? Also later, when you write “predictive confidence interval”, most people would call this a “predictive interval”. I think “confidence intervals” are about inference rather than prediction.

Q24: Figure 14: I think the results are very inconclusive, as the test coverage varies a lot, and not using the procedural improvement consistently performs better than using your procedural improvements in this metric, except for Aerial.

Q25: Where do you report the TLL for the image regression datasets?

Q26: Table 3: “For each problem, we highlight the best method in green, but we  also highlight in yellow methods with similar performance.” How do you define “best method” and how do you define “similar performance”? Do you have at least some vague rule of thumb to describe this?

Q27: Table 3: You should mention here that you are calibrating for validation coverage here.

Q28: Line 1459: training deep ensembles for 20k epochs is a lot, I think. How do you avoid overfitting? Do you select for each ensemble member the epoch where it had the lowest validation LL? Usually, a quite small number of epochs is selected, I guess?

Q29: Lines 1464-1465: “10000 epochs with an early stopping of 100 epochs”. What does “with an early stopping of 100 epochs” mean? Patience?

Q30: Line 1469: How do you calibrate your uncertainty if you use “all training data” for retraining?

Q31: Do you use any form of early stopping for the Image Regression Dataset?

My current score is ~5, and I am willing to change my score based on your answers. E.g., if you add the missing metrics.

---

> ### Author Response · Authors · 2025-11-28
> **Responses to weaknesses**
>
> **[W2]:  Different metrics across different datasets.**
>
> - Why have different metrics been used.
>
> We follow the protocol and existing literature for each dataset that we collected. Specifically, for the UCI regression datasets we adopted only RMSE and TLL computed using the predictive mean and total variance [1~4]. For the image regression datasets, we followed the metrics used in the paper [5], which focuses on evaluating the test coverage of models under distribution shift. This choice was made to ensure strict comparability with existing methods and to demonstrate that our approach performs well within the evaluation frameworks established by previous studies.
>
> - Unifying the metrics in the revised version.
>
> We agree that using different metrics across datasets may cause confusion and hurt the readability. In the revised version we adopt a unified evaluation protocol across all datasets. Specifically, we report the following five metrics: RMSE, TLL, WA, Test coverage, and Test interval length.
>
> - Reference
>
> 1. Skafte, N., Jørgensen, M., & Hauberg, S. (2019). Reliable training and estimation of variance networks. Advances in Neural Information Processing Systems, 32.
> 2. Seitzer, M., Tavakoli, A., Antic, D., & Martius, G. (2022) On the Pitfalls of Heteroscedastic Uncertainty Estimation with Probabilistic Neural Networks. In International Conference on Learning Representations.
> 3. Valdenegro-Toro, M., & Mori, D. S. (2022). A deeper look into aleatoric and epistemic uncertainty disentanglement. In 2022 IEEE/CVF Conference on Computer Vision and Pattern Recognition Workshops (CVPRW) (pp. 1508-1516). IEEE.
> 4. Immer, A., Palumbo, E., Marx, A., & Vogt, J. (2023). Effective bayesian heteroscedastic regression with deep neural networks. Advances in Neural Information Processing Systems, 36, 53996-54019.
> 5. Gustafsson, F. K., Danelljan, M., & Schön, T. B. (2023). How Reliable is Your Regression Model's Uncertainty Under Real-World Distribution Shifts?. Transactions on Machine Learning Research.
>
> **[W1, W3]: The performance varies across different datasets because the model lacks calibration.**
>
> Thank you. You are correct, except for the image regression datasets where we followed the original paper and adopted an additive calibration scheme.
>
> To address this concern, in the revised version we adopt a consistent and principled calibration strategy across all public datasets (UCI and image regression) following (https://arxiv.org/abs/1706.04599). Specifically, we use a constant c to calibrate on the predicted total variance on the validation datasets, and then report the final accuracy metrics with the calibrated total uncertainty for all compared methods. This actually further strengthened the performance of the proposed VeBNN method in the revised manuscript. Thank you!
>
> We also want to clarify our metrics choice for the plasticity-law datasets. Since this dataset provides test dataset with ground-truth aleatoric uncertainty via 100 repeated simulations per input, it allows a direct assessment of aleatoric accuracy using the WA. Therefore, using calibration would artificially distort WA and eliminate its interpretability. In addition, we wish to preserve the intrinsic behavior of the epistemic uncertainty, where it tightens as more data are observed. Therefore, we keep the calibration factor fixed to 1 for all methods, ensuring that WA and Epistemic TLL calculated based on ground-truth mean and predicted epistemic uncertainty reflect the raw predictive behavior of each model.  Furthermore, we will provide the test coverage and test interval length using the ground truth mean and predicted epistemic uncertainty for this dataset in the appendix at the final version for consistency of the metrics.
>
> **[W3]: Prior over network parameters.**
>
> We agree with the reviewer that the prior over neural parameters can strongly influence the scale of epistemic uncertainty. To ensure a fair and controlled comparison across all methods evaluated in this paper, we adopt a unit-variance Gaussian prior for all network parameters.
>
> **[W4, Q14]: Incorrect derivation of the Gamma log likelihood loss function.**
>
> We have re-examined both the analytical derivation and the implementation in our code. We identified a typo in the manuscript: the term involving r was incorrectly placed in the denominator, and you used the correct Gamma negative log-likelihood expression. In the revised version, we have corrected this typo by updating Eqs. (6) and (7). Thank you.
>
> **[W5]: Potential of using deep ensemble as inference method.**
>
> Good point. We have now also included ensembling as another option for epistemic uncertainty estimation of the VeBNN. This also helps illustrate that even for this strong baseline, the cooperative strategy also improves the estimates when compared to joint training with deep ensembles.

---

> ### Author Response · Authors · 2025-11-28
> **Response to questions**
>
> - [Q1]: We agreed. We changed “unacceptable” to “undesired” in the revised version.
> - [Q2]: Our intention was to explain that, in active learning, selecting the point with the largest epistemic uncertainty is not ideal for observations with heteroscedastic noise. Without uncertainty disentanglement, such a strategy may select points with high aleatoric uncertainty, reducing efficiency. We have clarified this explanation in the revised version.
> - [Q3]: we had already cited Lakshminarayanan et al. (2017), but now we include it in the suggested location too.
> - [Q4]: We replaced “has shown” into “suggests” as the statement reflects empirical results.
> - [Q5]: We switched those two notations in the revised version.
> - [Q6]: We deleted “with” in the revised version.
> - [Q7]: Once the BNN inference is executed, we use the posterior mean for Step 3. Otherwise, we use the MAP from Step 1.
> - [Q8]: We changed $\theta^*$ into $\Theta$ in the revised version, which represents the posterior distribution of $\theta$.
> - [Q9]: Yes.
> - [Q10]: We changed into “There are, however, multiple important details”
> - [Q11]: Thanks, we modified it to “the variance network estimates the distribution of the squared residuals”.
> - [Q12]: We added “as explained in Lemma 3.1.” at the end of the sentence.
> - [Q13]: We added a lemma before the proof.
> - [Q14]: See the answer of W4
> - [Q15]: Our original phrasing was unclear. what we meant is, the likelihood in our setting is Gaussian  and the prior over parameters is also Gaussian. Therefore, pSGLD provides a Gaussian-like approximate posterior according to Bayesian model average. We rephrased this sentence to state this more precisely and avoid confusion.
> - [Q16]: “TLL” refers to test log likelihood, we added the full term in the revised version
> - [Q17]: Thanks.
> - [Q18]: We updated the results according to your suggestion and enrich the discussion in the revised version based on the new results.
> - [Q19]: Yes, you are correct. Deep Ensembles benefit from parallelization. Still, there are two points to keep in mind. First, Deep Ensembles obtain one sample per run (even if the number of epochs was smaller). Second, we now also show that the cooperative strategy can be applied using ensembles at Step 3, and that also leads to improvements. So, we really believe that our arguments hold across the board. It’s not so much about performance, but about generality, practicality and method robustness.
> - [Q20]: Thanks.
> - [Q21]: 800 refers to the number of training sequences. We have clarified this in the revised version.
> - [Q22]: We only use validation data for the image regression datasets. Following the reviewer’s comments, we unified the evaluation metrics and the protocol across all methods, including the use of validation data for both UCI regression and image regression datasets.
> - [Q23]: We replaced “confidence interval” with “predictive interval.”
> - [Q24]: This has been updated accordingly with the new metrics.
> - [Q25~27]: Thanks, we have updated this table with new metrics that including TLL and proper calibration.
> - [Q28]: Since the number of required epochs for Deep Ensembles will differ across datasets, we use a sufficiently large number of epochs so that the method will work consistently across all datasets. In addition, we shuffle the training data so that part of it will be used for training and part for “validation” to select the best ensemble member.
> - [Q29]: Yes, it means patience.
> - [Q30]: This setup is taken from existing papers (https://arxiv.org/abs/1906.03260,  https://arxiv.org/abs/2203.09168, ). They simply validate the training epoch and learning rate and then retain the model with those hyperparameters. We adopt this setup directly.
> - [Q31]: No, we also took the training configuration for deep ensemble from paper (https://arxiv.org/abs/2302.03679). We will clarify this in the revised paper more carefully.

---

### Official Review · Reviewer_J3sN · 2025-11-01

**Soundness:** 2
**Presentation:** 3
**Contribution:** 3
**Rating:** 4
**Confidence:** 4

**Summary:**

This paper presents a cooperative training strategy to jointly estimate *aleatoric* and *epistemic* uncertainties in regression problems. The authors argue that mean–variance networks (MVEs) poorly estimate heteroscedastic variance and that standard Bayesian neural networks (BNNs) are unstable when directly modeling aleatoric uncertainty.

The proposed method, **VeBNN (Variance-estimating BNN)**, introduces a two-stage iterative scheme:
1. Train a deterministic mean network.
2. Train a **variance network** on squared residuals using a **Gamma likelihood**, theoretically justified under Gaussian noise.
3. Freeze the variance and perform Bayesian sampling (via pSGLD) to model epistemic uncertainty.

This cooperative process repeats for \(K\) iterations, selecting the best parameters through an estimated log marginal likelihood. The Gamma residual modeling provides numerical stability and better calibration.

Experiments include **UCI regression benchmarks**, **image-based regression under distribution shift**, and a **new physics-based dataset (material plasticity)** with *known aleatoric uncertainty*, enabling quantitative disentanglement analysis.

**Strengths:**

- **Mathematical rigor:** Correct derivation of Gamma residual likelihood; strong connection to heteroscedastic regression theory.
- **Stability:** Cooperative schedule avoids the gradient imbalance issues of Gaussian NLL.
- **Empirical depth:** Includes a dataset with known aleatoric uncertainty, allowing quantitative disentanglement.
- **Reproducibility:** Clear training details and ablations; implementation reproducible.
- **Trustworthy AI relevance:** Promotes calibrated and interpretable uncertainty estimation, valuable for safe AI systems.

**Weaknesses:**

1. **Assumption of unimodality:** The Gamma residual model presumes Gaussian, symmetric residuals. It may not generalize to multi-modal or skewed \(p(y|x)\).
   *Suggestion:* Add stress tests under Student-t or mixture noise.

2. **Incomplete baselines:** The study omits Mixture Density Networks (MDN) and Mixture-of-Experts (MoE), which directly model multi-modal uncertainties.

3. **No convergence analysis:** The cooperative iterations (\(K\)) are empirically motivated but lack theoretical justification.

4. **Limited empirical diversity:** Datasets are appropriate but not exhaustive—no explicit OOD or multi-modal scenarios tested.

5. **Moderate novelty:** The work refines existing ideas rather than introducing a new learning paradigm.

**Questions:**

1. How does the Gamma residual assumption behave under **non-Gaussian or multi-modal noise**?
2. Do multiple cooperative cycles (\(K>1\)) provably improve marginal likelihood, or is convergence purely empirical?
3. Would the results hold using other inference methods (e.g., VI, SGHMC) instead of pSGLD?
4. Have you measured **epistemic calibration** or coverage (ECE, TLL) under domain shift?
5. Will the **plasticity dataset** with known aleatoric uncertainty be released publicly for reproducibility?

---

> ### Author Response · Authors · 2025-11-28
> **Responses to weaknesses**
>
> **[W1, W2, W4, Q1]: Extension to multi-modal aleatoric noise; missing MDN/MoE baselines.**
>
> Thank you for this interesting suggestion. We implemented our cooperative strategy with Mixture Density Networks (MDN), enabling the model to handle multi-modal aleatoric noise. This integration is straightforward within our proposed framework: we simply replace the unimodal mean network with an MDN-based mean network and adjust the variance network accordingly. With this extension, our method (VeBNN) can represent a broader class of predictive distributions.
>
> We evaluated this MDN-based variant using the illustrative multi-modal example from Harakeh et al. [6], and we have included the results and discussion in the revised manuscript. This demonstrates that our cooperative framework naturally generalizes to multi-modal settings without requiring architectural changes to the Bayesian (epistemic) component.
>
> **References**
>
> 1. Skafte, N., Jørgensen, M., & Hauberg, S. (2019). Reliable training and estimation of variance networks. NeurIPS.
> 2. Seitzer, M., Tavakoli, A., Antic, D., & Martius, G. (2022). On the Pitfalls of Heteroscedastic Uncertainty Estimation with Probabilistic Neural Networks. ICLR.
> 3. Gustafsson, F. K., Danelljan, M., & Schön, T. B. (2023). How Reliable is Your Regression Model's Uncertainty Under Real-World Distribution Shifts? TMLR.
> 4. Valdenegro-Toro, M., & Mori, D. S. (2022). A deeper look into aleatoric and epistemic uncertainty disentanglement. CVPRW.
> 5. Immer, A., Palumbo, E., Marx, A., & Vogt, J. (2023). Effective Bayesian heteroscedastic regression with deep neural networks. NeurIPS.
> 6. Harakeh, A., Hu, J., Guan, N., Waslander, S., & Paull, L. (2023). Estimating regression predictive distributions with sample networks. AAAI.
>
> ---
> **[W3, Q2]: Theoretical improvement with cooperative cycles K>1.**
>
> The empirical evidence of improvement with cooperative cycles is strong, given that we considered 18 datasets and the improvement was observed in all of them (e.g., Figures 4 and 18). Prompted by this question, we worked on theoretical arguments to justify these observations -- see Appendix A.2 and A.3 (apologies, but it tooks us more than 3 pages). We start in Apppendix A.2 by analyzing the Hessian of the last layer of the MVE network, from where we found that its determinant is negative, i.e. the loss function has a saddle shape in the output space that explains the “trade-off” between the mean and aleatoric variance estimation when jointly training the MVE network (illustrated in Fig. 1). Furthermore, in Apppendix A.3, we use the finite-sample bias theorem to show that both cooperative and joint training of the MVE network lead to a bias error of the mean estimate, but due to the lack of convexity of the loss function in the case of joint training, its bias is larger. In addition, and specifically addressing the point you raised, we analyzed why a cooperative cycle leads to convergence by going through the two iterations (the argument holds for the next ones). In essence, iteration zero leads to an estimation of the aleatoric uncertainty that is necessarily better than the constant uncertainty assumed in Step 1 at iteration 0 because in the worst case scenario it returns the same constant variance (if that was correct). Therefore, the next iteration of Step 1 includes an updated aleatoric uncertainty estimate that tightens the mean prediction for points with lower uncertainty and relaxes it for points of high uncertainty, improving the mean prediction. The improvement in the mean prediction implies the subsequent improvement of the aleatoric variance in Step 2 because the mean is fixed.
>
> Finally, we also added a new analysis monitoring the validation log-likelihood over each step with K=5. We compare two scenarios:
> 1. using a properly trained ME network from Step 1, and
> 2. using an untrained ME network at Step 1.
>
> In both instances we observe convergence, although the latter requires a larger K.
>
> ---
> **[W5]: Moderate novelty**
>
> We understand this criticism, especially given that regression tasks are undeniably an old topic in Computer Science. However, your encouragement to look for theoretical arguments signifcantly strengthen this work, and show that this might become a long lasting result. We also want to highlight that we tested the method for feedforward neural networks, convolutional and recurrent neural networks. The method is truly robust, and we observe improvements compared to joint training for all types of inference that we could test, even when considering deep ensembles (i.e., Step 3 in the cooperative training when using deep ensembles leads to better performance compared to joint training).

---

> ### Author Response · Authors · 2025-11-28
> **Responses to questions**
>
> **[Q3].  Do the results hold using other inference methods?**
>
> Yes, VeBNN is compatible with different inference methods. We have evaluated on several inference approaches, BBB, MC Dropout, and pSGLD. VeBNN has significant improvement under the same inference approach across all datasets. We also tried SGLD. We already started these tests, and the conclusions hold so far, but we prioritized other recommendations.
>
> **[Q4]. Epistemic calibration or coverage (ECE, TLL) under domain shift?**
>
> We had already validated this in Appendix D.3. The image regression datasets are designated in the literature for evaluating uncertainty under domain shift. In the current version, we examined the test coverage with uncertainty calibrated by validation data as introduced by Eqs. (32)~(35).
>
> **[Q5]. Will the plasticity datasets be public.**
> Yes. The purpose of developing the plasticity dataset is not only to evaluate VeBNN but also to make the datasets publicly available for the broader AI and scientific modeling community. We will release all data as well as the implementation of the VeBNN method upon acceptance.

---

### Author Response · Authors · 2025-12-04
**Summary for the AC**

We thank all four reviewers for their detailed feedback. In the revised manuscript, we addressed every major concern for each reviewer:

- `Reviewer J3sN`: Added multi-modal experiments via an MDN-based extension; expanded theoretical justification for cooperative cycles (new Appendix A.2–A.3); clarified domain-shift evaluation and inference-method generality.

- `Reviewer aT67`: Added missing metrics (TLL/VLL, coverage, interval length) for all tested datasets, clarified calibration strategy, corrected the Gamma NLL derivation, and included results for ensembles under our cooperative scheme.  All the refinements and modifications lead to a thorough update on our experiments, which brings more significant improvement against SOTA methods.

- `Reviewer Zmnw` Clarified motivation, novelty, and positioning; improved presentation and added ablations.

- `Reviewer gD7g`: Improved explanation of disentanglement, added visualizations, and additional comparisons with the suggested baseline from Immer et al. (2023).

Together, these changes substantially strengthen both the theoretical foundation and the empirical evidence for VeBNN.

- Regarding the timing:
We intended to prepare our revisions carefully, as several of the reviewers’ suggestions (e.g., multi-modal extension, calibration study) required running substantial new experiments. Unfortunately, the unexpected and severe platform leakage incident occurred while we were still processing these results. Because of that interruption, we could no longer continue the discussion with reviewers or upload the revised material in time.

---

### Meta-Review · Area_Chair_qixv · 2025-12-24

**Summary:**

The reviewers have unanimously identified several serious concerns with the paper and have raised numerous questions and suggestions for improvement. The sheer number and scope of the required changes would necessitate a longer review process with extended back-and-forth discussions and evaluations, which is not feasible within the limited discussion period.

Moreover, while the paper aims to address an important challenge, the underlying ideas of the paper represent only marginal and incremental novelty relative to prior work. Although incorporating the reviewers’ suggestions could strengthen the presentation and sharpen the contributions, the core ideas largely amount to a straightforward adaptation and combination of existing approaches. As a result, the paper is not suitable for publication in its current form.

**Reviewer Concerns:**

While the authors have attempted to address several of the concerns raised in the rebuttal, e.g., adding new experiments, sharpening the presentation, and improving the justification of the methodology, the core issues remain, namely the limited novelty and incremental nature of the contributions.

**Reviewer Scores:**

Given the inherent issues raised by the reviewers (as well as the AC), such as the limited novelty and incremental nature of the contributions, and the fact that all reviewers initially rated the paper low, I believe that the revisions and rebuttal are very unlikely to lead most reviewers to substantially increase their scores, with most either moving to, or remaining at, a position marginally below the acceptance threshold.

---

### Decision · Program_Chairs · 2026-01-26

Reject